# Reasoning about mental states under uncertainty

Isu Cho¤a☯, Nellie Kamkar☯¤b*, Niki Hosseini-Kamkar¤c

Department of Psychology and Brain and Mind Institute, Western University, London, Ontario, Canada

☯ These authors contributed equally to this work.
¤a Current address: Department of Psychology, Brandeis University, Waltham, Massachusetts, United States of America
¤b Current address: Lawson Health Research Institute, London, Ontario, Canada
¤c Current address: Department of Psychiatry, McGill University, Montreal, Quebec, Canada
* nellie.kamkar@lhsc.on.ca

**Data Availability Statement:** The raw data were not institutionally approved by the Non-Medical Research Ethics Board to be shared on a repository. Data requests can be available upon request from the corresponding author. Preregistration details are publicly available at https://osf.io/89wr4/.

## Abstract

Theory of Mind (ToM), the ability to infer the mental states of others, is integral to facilitating healthy social interactions. People can reason about the mental states of others even with limited or (sometimes) inconsistent information. However, little is known about *how* people make inferences about the mental states of others under uncertainty, and what features of information are important in aiding mental state reasoning. In the current study, we conducted three unique experiments that alter participant's uncertainty when engaging in ToM tests. In Experiment 1, we simultaneously manipulated both the amount and consistency of information available in social stimuli presented to 59 participants. In Experiments 2 and 3, we aimed to decipher which feature of social stimuli is more conducive to mental state reasoning. Experiment 2 manipulated only the amount of information available to 47 participants, while Experiment 3 manipulated only the consistency of information available to 46 participants. Using both frequentist and Bayesian statistics, results confirmed that manipulating the amount and consistency of information alters ToM performance. Exploratory analysis comparing the effects of the amount and consistency of information suggests that the effects of the *consistency* of information seem to be stronger than those of the amount of information. Taken together, all three experiments suggest that while both the amount and consistency of information are important features of social stimuli—the consistency of information available is more salient when inferring mental states of others. These findings are discussed in relation to information theory and have important implications for creating enriched social stimuli, which may enhance mental state reasoning in individuals with social deficits.

## Introduction

A capacity to infer mental states, or theory of mind (ToM), is integral to facilitating our basic daily social and communicative interactions. Given the overarching importance of ToM in

**Funding:** The author(s) received no specific funding for this work.

**Competing interests:** The authors have declared that no competing interests exist.

social relations, many researchers have examined the development of ToM in children [1], and several methods of measuring ToM in healthy adults have been developed [2]. However, relatively little information is available on how adults integrate various information sources to reason about others' mental states. For example, a large body of research has provided instrumental insights on how ToM abilities can be measured (using a variety of ToM measurements) [2–4] and how they differ across individuals with different characteristics (e.g., language ability [5, 6], bilingualism [7–9], birth order [10], and socioeconomic status [11]). Little is known, however, about how healthy adults make inferences about the mental states of others. Specifically, there are complex sources of social information available in real-world interactions, and little empirical evidence is available that sheds light on the sources of information that are most important in integrating social cues to make mental state inferences.

In most of the previous research on ToM, participants are given limited information that must be used to infer the mental state of others (see the Strange Stories Film task [12] for exception). For example, in a false-belief task (such as the Sally-Anne task [13]), participants are given limited pieces of information (e.g., Sally put a marble in location A and did not see that Anne moved it to location B) that should correspond with a correct answer to a ToM question (e.g., Question: Where will Sally look for the marble? Answer: location A). When measuring ToM abilities, it is typical for researchers to provide limited information that corresponds to a correct mental state; however, it is challenging to explore *how* people make inferences about mental states (i.e., do they rely more on the amount or consistency of information provided to make a mental state inference?).

Furthermore, in real life, there are a variety of information sources that can be used to infer the mental states of others and sometimes, some of the information is inconsistent or contradicts the previous information obtained. We have the remarkable ability to integrate a broad variety of often incongruent and convoluted information together to infer the mental states of others. This warrants a deeper examination of how people make these mental-state inferences based on the complexity of information sources provided to them.

Thus, the goal of the present set of experiments is to explore, from a new direction, how ToM operates by manipulating the features of mental state-related information. Here, we use an information-theoretic perspective to better understand the mechanisms underlying adult's mental state reasoning abilities. An information-theoretic perspective to understanding ToM asserts that ToM processing generates a series of possible representations of another agent's mental state, each with varying levels of certainty [14]. To determine whether ToM operates through gradual uncertainty-reduction, we alter participants' certainty regarding an agent's mental state and measure their ability to reason about those mental states thereafter. In Experiment 1, participants' certainty levels regarding an agent's mental states were experimentally manipulated by altering the informational richness of the stimulus. Specifically, both the amount and consistency of information provided to participants were altered and participants' responses to reason about the agent's mental states were analyzed. We predict that altering one's uncertainty about an agent's mental states will have a direct impact on scores on a test of ToM (please see the design section in Experiment 1 for a detailed explanation on the definition of congruency score). We predict that performance will increase as one's uncertainty about the agent's mental state decreases. In two follow-up experiments, we aimed to discern whether the amount of information alone, the consistency of information alone, or both are critical in altering mental state attributions. In Experiment 2, we aimed to determine whether specifically and selectively varying only the *amount—or* quantity—of information will influence ToM. We predict that manipulating the quantity of information will have a direct effect on ToM performance such that as the amount of information provided increases, congruency scores on ToM tests will also increase. In Experiment 3, we determined whether specifically and selectively

altering the consistency of information contributed to performance on a ToM task. Here, consistency of information was manipulated by providing participants with varying degrees of statements that were either consistent or contradictory with the agent's mental state. We predict that the consistency of information will directly influence performance on ToM, such that receiving more consistent information will yield higher congruency scores on the ToM tasks. Prior to collecting data, we preregistered our hypotheses, methods, and analysis plans for all three studies on the Open Science Framework (https://osf.io/89wr4/). Analyses conducted for all three studies involve *a priori*, hypothesis-driven analysis plans.

## Experiment 1

In order to make a mental state inference about the agent in the vignette, participants require information that is consistent with the set of mental state inferences determined at the outset of the trial (i.e., that Lewis likes Clara's piano playing). In Experiment 1, the number of relevant sentences was manipulated by inserting filler sentences that have nothing to do with the agent's mental states (i.e., that Lewis speaks four languages). In addition, the consistency of sentences was manipulated by inserting sentences that are directly opposite to the mental state inferences of the agent (i.e., Lewis wears noise-canceling headphones when Clara plays the piano). In this experiment, the independent variable was the degree of uncertainty with three levels—low, intermediate, and high uncertainty. Critically, uncertainty was manipulated using both the amount and consistency of information. The dependent variable was congruency scores (a continuous variable) on questions regarding an agent's mental state. We predict a direct effect of uncertainty manipulation on ToM performance such that as one's uncertainty regarding an agent's mental states decreases, their congruency scores on a test of ToM will also increase; whereas, as one's uncertainty regarding an agent's mental state increases, the congruency scores will decrease.

### Methods

**Ethics statement.**    Institutional Review Board (IRB) ethical approval was obtained by Western University's Research Ethics Board (REB) for non-medical research (file # 104311). All participants provided written informed consent prior to participating in the study.

**Participants.**    Of 59 participants who were tested, three were excluded from the analyses because they did not provide a complete dataset, leaving a final sample of 56 healthy adult participants (25 female) between 18 to 31 years of age ($M = 20.63$; $SD = 3.40$). Participants were undergraduate students at Western University selected from a university sample pool. Based on an *a priori* power analysis with 95% power, a medium effect size, and an alpha value of .05, we needed forty-three participants. Given that some participants might not meet the inclusion criteria, our stopping rule was to continue collecting data until the end of the academic semester during which the actual sample size exceeded the target sample size based on the power analysis. With 56 participants, we exceeded our sample size goal.

**Design.**    Experiment 1 was conducted using a within-subjects experimental design with 3 conditions. There was one within-subject independent variable, degree of uncertainty, with three levels: low, intermediate, and high. To address the hypothesis of a direct effect of the uncertainty manipulation on responses to questions pertaining to the agent's mental states (as a continuous dependent variable), a repeated-measures ANOVA was used (with follow-up pairwise tests if applicable). In addition to frequentist statistics, Bayesian statistical analyses (Bayes factor) were conducted to determine differences in participants' responses to the mental state inference questions across the three conditions. Trials began with a vignette that included a brief introduction followed by six sentences. In the intermediate-uncertainty

condition, participants received 2 sentences that provided relevant and consistent information about an agent's mental state, while the remaining four sentences served as length-matched fillers that ensure consistency across conditions in the amount of text displayed in each vignette. In the low-uncertainty condition, participants were given six sentences about behavior that provided relevant and consistent information which supported an inference about an agent's mental state(s) that we, as the experimenters, deemed the "correct" set of mental states. Thus, responses corresponding to this set of mental states were deemed as "congruent." In the high-uncertainty condition, participants were given three sentences with consistent information and three sentences with inconsistent information about an agent's mental states. Importantly, the information in the sentences included behaviours and not direct statements about the character's mental states. This way, we could ensure that participants were not simply memorizing and reporting back parts of the vignette but were attributing mental states based on the behaviour of the characters. The order of the six sentences on each trial was random. For an example trial from each condition, see **Table 1**.

On each trial, after participants read a vignette, they were shown a series of test questions assessing their ability to reason about the character's mental states. To comprehensively capture the constituent aspects of ToM, participants were asked questions that assessed their ability to reason about the character's beliefs, thoughts, knowledge, emotions, and intentions. For belief questions, the word "believes" was included in a true or false formatted question such as the following: "Lewis believes that Clara is musically talented". For questions about others' thoughts and knowledge, the words "thinks" and "knows" were included in a similar format. For questions evaluating participants' ability to reason about the emotions of others, the six basic emotions of happiness, sadness, anger, fear, surprise, and disgust [15] were used in multiple choice style questions such as the following: "when Lewis listens to Clara play the piano, how does he feel?" along with the six basic emotions as response options ranging from a) to f). And finally, for questions evaluating participants' ability to reason about the intentions of

**Table 1. Example vignette across the three conditions of Experiment 1.**

| Intermediate-Uncertainty Condition | Low-Uncertainty Condition | High-Uncertainty Condition |
|---|---|---|
| **Introduction**. Meet Lewis. Lewis has a roommate named Clara. Clara plays the piano. | **Introduction**. Meet Lewis. Lewis has a roommate named Clara. Clara plays the piano. | **Introduction**. Meet Lewis. Lewis has a roommate named Clara. Clara plays the piano. |
| **Filler 1**. Lewis speaks four languages because growing up, his father was a linguist who encouraged Lewis to learn multiple languages. | **Consistent 1**. Lewis spends most of his evenings listening to Clara play the piano. | **Consistent 1**. Lewis spends most of his evenings listening to Clara play the piano. |
| **Filler 2**. Lewis and Clara's apartment is downtown Chicago, which can get very expensive. Luckily their landlord covers the cost of utilities. | **Consistent 2**. After several hours of listening to her play, Lewis often smiles and applauds. | **Inconsistent 1**. After several hours of listening to her play, Lewis often gets a headache. |
| **Consistent 1**. Lewis spends most of his evenings listening to Clara play the piano. After several hours of listening to her play, Lewis often goes to the kitchen. | **Consistent 3**. When Clara's piano needed tuning, Lewis paid to have it done professionally. | **Inconsistent 2**. He once went to the mall across the street and bought a pair of noise-canceling headphones. |
| **Filler 3**. He then makes himself a cup of chamomile tea to help him fall asleep. | **Consistent 4**. Lewis regularly organizes social get-togethers in their apartment and asks Clara to play the piano for their friends. | **Consistent 2**. When Clara's piano needed tuning, Lewis paid to have it done professionally. |
| **Consistent 2**. One day, Lewis came across an advertisement to join the local orchestra. Lewis took a copy of the advertisement home and give it to Clara. | **Consistent 5**. He even once videotaped Clara playing the piano and uploaded the video on YouTube for the world to see. | **Inconsistent 3**. During a social get-together at their apartment, Clara played the piano and Lewis wore earplugs during her performance. |
| **Filler 4**. That night, Lewis's mother called and when they spoke, Lewis asked his mother if they would be visiting Chicago over the holidays. | **Consistent 6**. One day, Lewis came across an advertisement to join the local orchestra. Lewis took a copy of the advertisement home and gave it to Clara. | **Consistent 3**. One day, Lewis came across an advertisement to join the local orchestra. Lewis took a copy of the advertisement home and gave it to Clara. |

*Note*. Bolded and underlined text were not included in the text presented to participants

others, true or false questions such as the following were included: "if Clara joins the orchestra, Lewis will not purchase a ticket to her first concert".

Fifteen sets of vignettes involving different characters in various scenarios were created. Each set consisted of vignettes that came in three versions, one for each condition. On each trial, a vignette from a set that had not previously been selected was sampled without replacement. Vignettes were presented and responses were collected using OpenSesame© Experiment Builder 2.9.7 [16]. For details pertaining to the creation of the vignettes, please refer to the S1 and S2 Appendices. Condition order and sentence order were randomized; this is because there is evidence that condition order may be an important variable [17], thus in the present set of experiments, we controlled for condition order by randomizing the order of sentences. Specifically, the location of each sentence (i.e., a filler sentence, a consistent sentence or an inconsistent sentence) within the block of text for each vignette was random. In addition to this, whether the first vignette that a participant sees is an intermediate-uncertainty, low-uncertainty, or high-uncertainty condition was also randomized to ensure that we have controlled for all possible order effects. And finally, it is possible that some statements may be more potent and convincing in their content than others. For example, of the 6 "consistent" sentences, although all attempt to convey the notion that Lewis likes Clara's piano playing, it is possible that some may be stronger in portraying this mental state attribution than others and that there are individual differences in which statements participants feel convey more potent information on mental states. To control for this, sentences chosen from the 6 consistent sentences in the low-uncertainty condition to be inserted in the other conditions (intermediate-uncertainty and high-uncertainty) were randomly selected.

To obtain the continuous dependent variable of a congruency score for each condition from participants, the five responses to the test questions (believe, think, know, emotion, and intention) on each trial were aggregated—given five trials per condition, this resulted in a congruency score out of twenty-five. The "congruent" set of mental state attributions were determined *a priori* regardless of the type of condition (i.e., low/high/intermediate uncertainty conditions). Congruent scores were specified as those corresponding to the answers when only consistent information was provided. For example, at the outset of the trial, the experimenters pre-set a high "congruency score" to reflect scores indicating that the character Lewis likes Clara's piano playing. Thus, lower congruency scores in the high uncertainty condition do not indicate poor ToM ability, but rather, that participants' responses reflected the information provided, which is inconsistent with the pre-set "congruent" response. Our hypotheses pertained to the extent to which the amount and type of information that participants are given influences their final mental-state attributions.

Therefore, it is noteworthy that the term "congruency score" is not necessarily reflective of true correct responses, as the experimenters' pre-set list of "congruent" responses are by no means objectively true and accurate statements. That is, in conditions in which more "inconsistent" information is provided (i.e., that Lewis wears noise canceling headphones when Clara plays the piano), participants may respond in ways that indicate that Lewis *does not* like Clara's piano playing. While this response pattern is, in effect, correct, in the present set of Experiments the reference term of "congruency score" being low simply reflects the extent to which participants agree with the response options that have been predetermined *a priori*. This understanding of the congruency score is integral to interpreting the response options of participants in all three Experiments.

**Procedure.**   After providing written consent to participate in the study, participants were seated in front of a computer screen and told they would see a series of short stories ("paragraphs of text") on the screen. They were instructed to read each story as carefully as possible and were reassured that they would have ample time to read all the stories presented. They

were then told that once they were comfortable with their knowledge of each story, they could press any key on the keyboard to generate a series of questions pertaining to the story they just read. They were specifically told to focus their attention on "getting the answers right" rather than quickly reading and answering questions. Once they confirmed that they understood the instructions, the task started. Participants pressed the "T" or "F" keys to indicate whether the test questions were true or false, respectively. To answer Multiple Choice style questions, participants simply pressed the key corresponding to their choice (i.e., "A" to choose option a). Once finished, participants were given one full research course credit for their participation and were thanked for their time. The entire session took approximately 45 minutes to complete.

## Results and discussion

A repeated-measures ANOVA revealed a statistically significant effect of uncertainty condition on congruency scores (a continuous variable; see **Fig 1**), $F(1.79, 98.48) = 112.03$ $p < .001$, $\eta p^2 = .67$.

Follow-up pairwise comparisons revealed that all three conditions statistically differed from one another (**Table 2**): congruency scores were highest in the low-uncertainty condition ($M = 81.93$; $SD = 8.68$), moderate in the intermediate condition ($M = 74.98$;

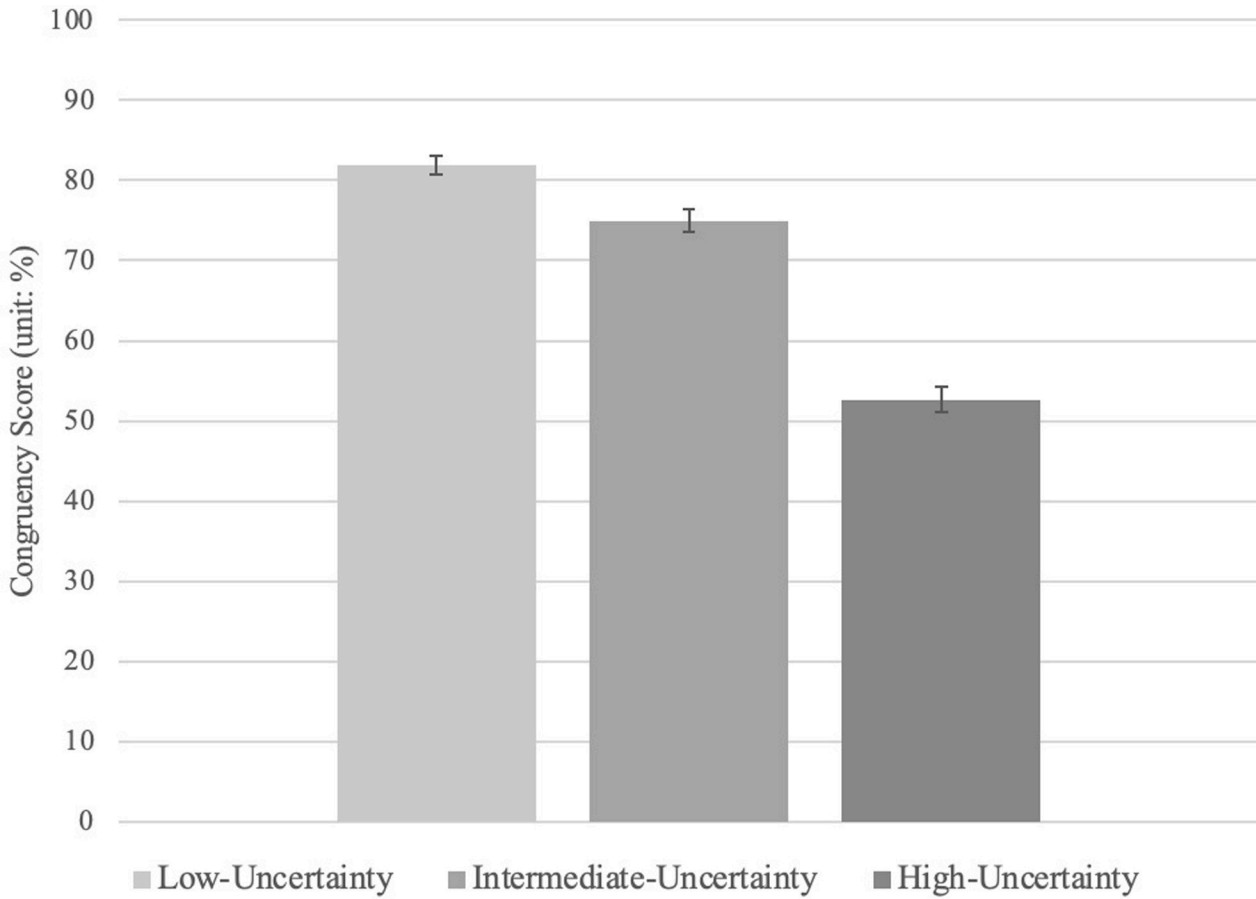

**Fig 1. Congruency scores as a function of uncertainty in Experiment 1.** All conditions differ at a significance level of $p < .001$. Error bars represent standard error. *Note. Congruency scores (%) reflect the degree to which responses correspond with the set of predetermined mental-states of the characters in the Vignettes *a priori*.

**Table 2. Results of the Bonferroni-corrected pairwise comparisons between the three conditions of Experiment 1.**

| Comparisons | Mean difference | 95% Confidence Interval for Difference [Lower, Upper] | p-value (Bonferroni-corrected) | BF$_{10,U}$* |
|---|---|---|---|---|
| Low vs. Intermediate Uncertainty | 6.95 | [2.95, 10.95] | < .001 | 297.20 |
| Intermediate vs. High Uncertainty | 22.36 | [16.65, 28.07] | < .001 | 4.30e+10 |
| Low vs. High Uncertainty | 29.30 | [24.01, 34.60] | < .001 | 2.09e+16 |

*Note.* BF$_{10,U}$* = Uncorrected Bayes Factor

$SD$ = 10.68), and lowest in the high-uncertainty condition ($M$ = 52.63; $SD$ = 11.96). We complemented these frequentist statistics with Bayesian statistics using JASP (JASP Team, 2018 [18]). The Bayes factors (with a default Cauchy prior width of 0.71) suggest that the data are more likely to be observed under the alternative hypotheses, relative to the null hypotheses, BF$_{10}$ = 1.12e +31—this is considered decisive evidence in favor the alternative [19]. Thus, both frequentist and Bayesian analyses suggest there is an effect of the uncertainty manipulation on performance.

In Experiment 1, we manipulated both the amount and consistency of information that participants were exposed to simultaneously. Relative to the intermediate-uncertainty condition, which included two relevant and consistent statements about of an agent's mental state and 4 statements that were irrelevant, the low-uncertainty condition increased the amount of information by providing 6 relevant and consistent statements. In contrast, the high-uncertainty condition provided 3 relevant and consistent statements but also 3 relevant but inconsistent statements. We found significant differences in participants' performance in congruency scores on test questions across all conditions.

Specifically, congruency scores were highest in the low-uncertainty condition, followed by the intermediate-uncertainty condition, and lowest in the high-uncertainty condition. Importantly however, because we manipulated both the amount and consistency of information simultaneously in Experiment 1, it is unclear if one manipulation (the amount or consistency of information provided) is more critical in influencing mental state attributions. Thus, in two follow-up experiments, we separately investigated the effect of each manipulation (amount and consistency of information) on participants' mental state attributions.

## Experiment 2

Like Experiment 1, for participants to make a mental state inference about the agent in the vignette, they require information that is consistent with the set of mental state inferences determined at the outset of the trial (i.e., that Lewis likes Clara's piano playing). In Experiment 2, only the number of relevant sentences was manipulated by inserting filler sentences that have nothing to do with the agent's mental states (i.e., that Lewis speaks four languages). In Experiment 2, we aimed to discern whether specifically and selectively altering only the *amount* of relevant information would alter participants' mental state attributions. We therefore altered participants' certainty regarding an agent's mental states by manipulating the *quantity* of information provided to them only—thus the independent variable was informational quantity with three levels, low, intermediate, and high. We then measured participants' ability to reason about an agents' mental states as a function of varying the amount of relevant information provided in the stimulus. From an information-theoretic perspective, it was predicted that as more relevant behavioral information is provided to support a mental state inference, congruency scores would increase.

## Methods

**Ethics statement.** IRB ethical approval was obtained by Western University's REB for non-medical research (file # 104311). All participants provided written informed consent prior to participating in the study.

**Participants.** Forty-seven healthy adult participants (37 females) between 17 to 23 years of age ($M = 18.45$; $SD = 1.21$) were tested in Experiment 2. Participants were undergraduate students at Western University selected from a university sample pool. No participants were excluded from the final analyses, as all participants provided complete data. Like Experiment 1, prior to collecting any data, we conducted a power analysis using G*Power 3.1. Based on our power analysis with 95% power, a medium effect size, and an alpha value of .05, we needed 43 participants. Given that some participants might not meet the inclusion criteria, our stopping rule was to continue collecting data until the end of the academic semester. With 47 participants, we exceeded our sample size goal.

**Design.** The design in Experiment 2 was similar to Experiment 1 with the following modifications. Experiment 2 was conducted using a within-subjects experimental design with 3 conditions. In the intermediate-quantity condition, four of the six total sentences provided relevant information about the mental states of an agent. The remaining two sentences were irrelevant filler sentences. In the high-quantity condition, all six sentences that participants were provided with were consistent with mental state(s). This condition provided participants with the highest amount of information to support mental state inferences. In the low-quantity condition, only two of the six total sentences were consistent with the character's "congruent" set of mental states (i.e., the mental states corresponding to the consistent information), with the majority of sentences (the remaining four) consisting of irrelevant filler sentences. To address the hypothesis of a direct effect of manipulating uncertainty via direct alterations of the quantity of consistent sentences, a repeated-measures ANOVA was used (with follow-up pairwise tests if applicable). In addition to frequentist statistics, Bayesian statistical analyses (Bayes factor) were conducted to determine differences in participant's responses to the mental state inference questions across the three conditions. See **Table 3** for an example of all conditions in Experiment 2.

**Procedure.** The procedure of Experiment 2 was identical to that of Experiment 1.

## Results and discussion

A repeated-measures ANOVA revealed that there is a statistically significant effect of condition on congruency scores (a continuous variable; see **Fig 2**), $F(2, 92) = 7.47$, $p < .001$, $\eta p^2 = .14$.

The Bayes factors show a similar pattern of results—that the data are more likely to be observed under the alternative hypotheses relative to the null hypotheses, $BF_{10} = 41.02$. Follow-up pairwise comparisons are provided in **Table 4**. Congruency scores in the intermediate-quantity condition ($M = 79.47$; $SD = 9.10$) were not significantly different than those in the high-quantity condition ($M = 81.04$; $SD = 10.44$). However, all other conditions significantly differed from one another, such that congruency scores were lower in the low-quantity condition ($M = 74.06$; $SD = 11.06$) than either the intermediate- or high-quantity conditions.

In Experiment 2, we manipulated only the amount of information that participants were exposed to. Participants received 6 informative and 0 uninformative (high-quantity), 4 informative and 2 uninformative filler (intermediate-quantity), or 2 informative and 4 uninformative filler (low-quantity) statements about an agent's mental states. We found significant differences in participants' congruency scores on test questions between the low-quantity condition and the intermediate and high-quantity conditions. This indicates that the low-quantity condition, in which an impoverished amount of information was provided to participants,

**Table 3. Example vignettes across the three conditions of Experiment 2.**

| Intermediate-Quantity Condition | High-Quantity Condition | Low-Quantity Condition |
|---|---|---|
| **Introduction**. Meet Lewis. Lewis has a roommate named Clara. Clara plays the piano. | **Introduction**. Meet Lewis. Lewis has a roommate named Clara. Clara plays the piano. | **Introduction**. Meet Lewis. Lewis has a roommate named Clara. Clara plays the piano. |
| **Filler 1**. Lewis and Clara's apartment is downtown Chicago, which can get very expensive. Luckily their landlord covers the cost of utilities. | **Consistent 1**. Lewis spends most of his evenings listening to Clara play the piano. | **Filler 1**. Lewis speaks four languages because growing up, his father was a linguist who encouraged Lewis to learn multiple languages. |
| **Consistent 1**. Lewis spends most of his evenings listening to Clara play the piano. | **Consistent 2**. After several hours of listening to her play, Lewis often smiles and applauds. | **Filler 2**. Lewis and Clara's apartment is downtown Chicago, which can get very expensive. Luckily, their landlord covers the cost of utilities. |
| **Consistent 2**. After several hours of listening to her play, Lewis often smiles and applauds. | **Consistent 3**. When Clara's piano needed tuning, Lewis paid to have it done professionally. | **Consistent 1**. Lewis spends most of his evenings listening to Clara play the piano. After several hours of listening to her play, Lewis often goes to the kitchen. |
| **Consistent 3**. He even once videotaped Clara playing the piano and uploaded the video on YouTube for the world to see. | **Consistent 4**. Lewis regularly organizes social get-togethers in their apartment and asks Clara to play the piano for their friends. | **Filler 3**. He then makes himself a cup of chamomile tea to help him fall asleep. |
| **Consistent 4**. One day, Lewis came across an advertisement to join the local orchestra. Lewis took a copy of the advertisement home and gave it to Clara | **Consistent 5**. He even once videotaped Clara playing the piano and uploaded the video on YouTube for the world to see. | **Consistent 2**. One day, Lewis came across an advertisement to join the local orchestra. Lewis took a copy of the advertisement home and gave it to Clara. |
| **Filler 2**. That night, Lewis's mother called and when they spoke, Lewis asked his mother if they would be visiting Chicago over the holidays. | **Consistent 6**. One day, Lewis came across an advertisement to join the local orchestra. Lewis took a copy of the advertisement home and gave it to Clara. | **Filler 4**. That night, Lewis's mother called and when they spoke, Lewis asked his mother if they would be visiting Chicago over the holidays. |

*Note*. Bolded and underlined text were not included in the text presented to participants.

reduced their congruency scores. Congruency scores did not significantly differ, however, between the intermediate-quantity condition and the high-quantity condition. These results are similar to those of Experiment 1, with the exception that the intermediate-quantity condition was not any more difficult than the high-quantity condition. Further interpretations of these findings are provided in the General Discussion.

## Experiment 3

Like Experiments 1 and 2, to make mental state inferences that are in line with those pre-set at the outset of the trial, participants require information that is consistent (i.e., that Lewis likes Clara's piano playing). In Experiment 3, only the consistency of sentences was manipulated by inserting sentences that are directly opposite to the pre-set mental state inference (i.e., that Lewis wears noise-canceling headphones when Clara plays the piano). In Experiment 3, we aimed to discern whether specifically and selectively altering only the *consistency* of relevant information would alter participants' mental state attributions. We therefore measured participants' ability to reason about an agents' mental states by manipulating the *consistency* of information provided to them only. Here, the independent variable was consistency of information, with three levels, low, intermediate, and high. From an information-theoretic perspective, it was predicted that as the behavioral information provided becomes consistent with supporting a mental state inference, congruency scores in mental state reasoning would increase. Like Experiments 1 and 2, Experiment 3 was conducted using a within-subjects experimental design with 3 conditions. To address the hypothesis of a direct effect of manipulating uncertainty via direct alterations of the consistency of statements (i.e., including inconsistent statements), a repeated-measures ANOVA was used (with follow-up pairwise tests if applicable). In addition to frequentist statistics, Bayesian statistical analyses (Bayes factor) were conducted to determine differences in participant's responses to the mental state inference questions across the three conditions.

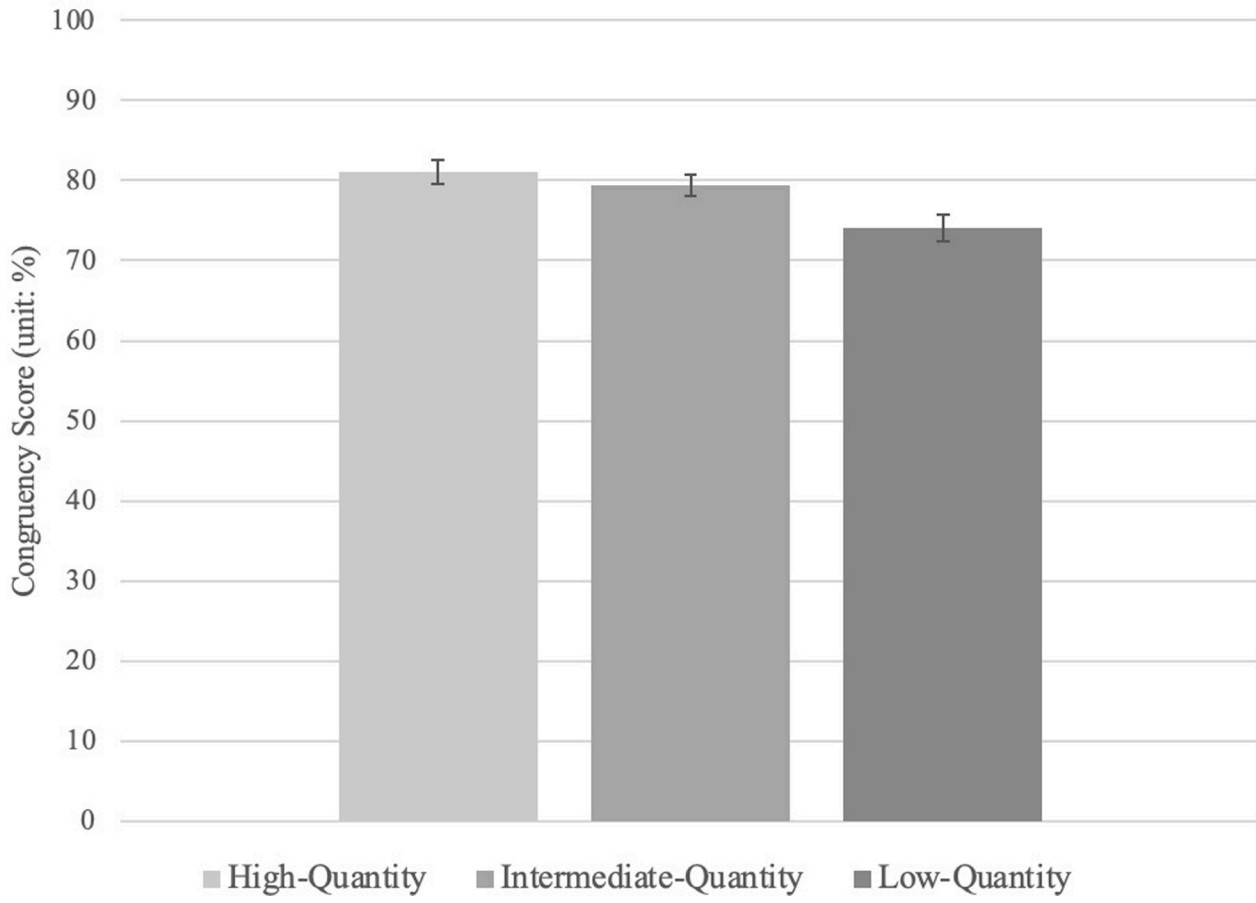

**Fig 2. Congruency scores as a function of uncertainty in Experiment 2.** Error bars represent standard error. *Note. Congruency scores (%) reflect the degree to which responses correspond with the set of predetermined mental-states of the characters in the Vignettes *a priori*.

## Methods

**Ethics statement.** IRB ethical approval was obtained by Western University's REB for non-medical research (file # 104311). All participants provided written informed consent prior to participating in the study.

**Participants.** Forty-six healthy adult participants (26 female) were tested in Experiment 3, who ranged in age from 17 to 28 years of age ($M = 18.53$; $SD = 2.27$). Participants were undergraduate students at Western University selected from a university sample pool. Like Experiment 1 and Experiment 2, prior to collecting any data, we conducted a power analysis using $G^*Power\ 3.1$. Based on our power analysis with 95% power, a medium effect size, and an alpha

**Table 4. Results of the Bonferroni-corrected pairwise comparisons between the three conditions of Experiment 2.**

| Comparisons | Mean difference | 95% Confidence Interval for Difference [Lower, Upper] | p value (Bonferroni-corrected) | BF$_{10,U}$* |
|---|---|---|---|---|
| High-Quantity vs. Intermediate-Quantity | 1.57 | [-2.48, 5.63] | 1.000 | 0.25 |
| Intermediate-Quantity vs. Low-Quantity | 5.40 | [0.82, 9.99] | .016 | 6.71 |
| High-Quantity vs. Low-Quantity | 6.98 | [1.59, 12.37] | .007 | 13.55 |

*Note.* BF$_{10,U}$* = Uncorrected Bayes Factor

value of .05, we needed forty-three participants; with 46 participants, we exceeded our sample size goal.

**Design.** The design in Experiment 3 was similar to that of Experiment 1 and Experiment 2 with the following modifications. In the intermediate-consistency condition, four of the six total sentences provided relevant information about the mental states of an agent. The remaining two sentences were contradictory sentences providing information that was inconsistent with the agent's mental states. Like Experiment 1 and Experiment 2, in the high-consistency condition, all six sentences that participants were provided with were consistent with mental state(s). This condition provided participants with the highest amount of information to support mental state inferences. In the low-consistency condition, only two of the six total sentences were consistent with the character's "congruent" set of mental states, with the majority of sentences (the remaining four) composed of contradictory inconsistent sentences. See **Table 5** for an example of all conditions in Experiment 3.

**Procedure.** The procedure of Experiment 3 was identical to that of Experiment 1 and Experiment 2.

## Results and discussion

A repeated-measures ANOVA revealed that there is a statistically significant effect of condition on congruency scores (a continuous variable; see **Fig 3**), $F(1.62, 73.00) = 187.82$, $p = .000$, $\eta p^2 = .81$.

Follow-up pairwise comparisons revealed that all three conditions statistically differed from one another (**Table 6**): congruency scores were highest in the high-consistency condition ($M = 83.59$; $SD = 7.95$), followed by the intermediate-consistency condition ($M = 60.72$; $SD = 15.76$), and lowest in the low-consistency condition ($M = 30.78$; $SD = 12.43$). Bayesian analysis show that the data were more likely to be observed under the alternative hypotheses relative to the null hypotheses, $BF_{10} = 1.16e +40$.

**Table 5. Example vignettes across the three conditions of Experiment 3.**

| Intermediate-Consistency Condition | High-Consistency Condition | Low-Consistency Condition |
|---|---|---|
| **Introduction**. Meet Lewis. Lewis has a roommate named Clara. Clara plays the piano. | **Introduction**. Meet Lewis. Lewis has a roommate named Clara. Clara plays the piano. | **Introduction**. Meet Lewis. Lewis has a has a roommate named Clara. Clara plays the piano. |
| **Consistent 1**. Lewis spends most of his evenings listening to Clara play the piano. | **Consistent 1**. Lewis spends most of his evenings listening to Clara play the piano. | **Consistent 1**. Lewis spends most of his evenings listening to Clara play the piano. |
| **Consistent 2**. After several hours of listening to her play, Lewis often smiles and applauds. | **Consistent 2**. After several hours of listening to her play, Lewis often smiles and applauds. | **Consistent 2**. One day, Lewis came across an advertisement to join the local orchestra. Lewis took a copy of the advertisement home and gave it to Clara. |
| **Inconsistent/Contradictory 1**. He once went to the mall across the street and bought a pair of noise-canceling headphones. | **Consistent 3**. When Clara's piano needed tuning, Lewis paid to have it done professionally. | **Inconsistent/Contradictory 1**. After several hours of listening to her play, Lewis often gets a headache. |
| **Consistent 3**. He even once videotaped Clara playing the piano and uploaded the video on YouTube for the world to see. | **Consistent 4**. Lewis regularly organizes social get-togethers in their apartment and asks Clara to play the piano for their friends. | **Inconsistent/Contradictory 2**. He once went to the mall across the street and bought a pair of noise-canceling headphones. |
| **Inconsistent/Contradictory 2**. During a social get-together at their apartment, Clara played the piano and Lewis wore earplugs during her performance. | **Consistent 5**. He even once videotaped Clara playing the piano and uploaded the video on YouTube for the world to see. | **Inconsistent/Contradictory 3**. During a social get-together at their apartment, Clara played the piano and Lewis wore earplugs during her performance. |
| **Consistent 4**. One day, Lewis came across an advertisement to join the local orchestra. Lewis took a copy of the advertisement home and gave it to Clara. | **Consistent 6**. One day, Lewis came across an advertisement to join the local orchestra. Lewis took a copy of the advertisement home and gave it to Clara. | **Inconsistent/Contradictory 4**. When one of their mutual friends asked Lewis if Clara should play the piano at their wedding, Lewis advised him against it. |

*Note*. Bolded and underlined text were not included in the text presented to participants.

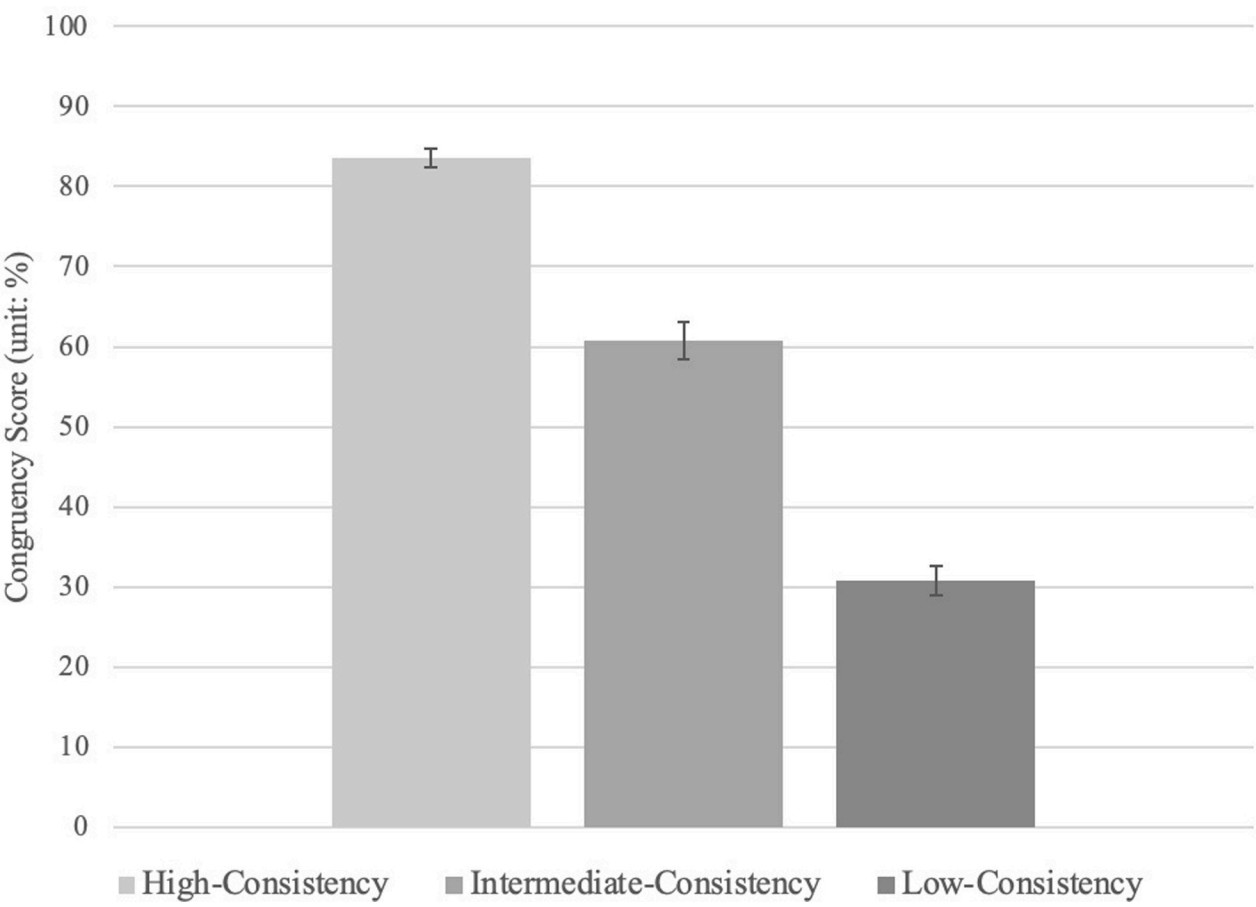

**Fig 3. Congruency scores as a function of uncertainty in Experiment 3.** All conditions differ at a significance level of $p < .001$. Error bars represent standard error. *Note. Congruency scores (%) reflect the degree to which responses correspond with the set of predetermined mental-states of the characters in the Vignettes *a priori*.

In Experiment 3, we provided the same amount of mental state-relevant information (i.e., 6 sentences) but manipulated only the *consistency* of information that participants were exposed to. Participants received 6 consistent and 0 contradictory (high-consistency), 4 consistent and 2 contradictory (intermediate-consistency), or 2 consistent and 4 contradictory (low-consistency) statements about an agent's mental states. We found significant differences in participants' congruency scores on test questions across all conditions. As predicted, performance in congruency scores was highest in the high-consistency condition, followed by the intermediate-consistency condition, and lowest in the low-consistency condition. The results of Experiment 3 show strong and consistent patterns indicating that selective manipulations of the consistency of information that participants have access to significantly alters their patterns of mental state reasoning.

### Exploratory analysis comparing the intermediate condition of Experiments 2 and 3

In addition to the above analyses, we conducted an exploratory analysis comparing the intermediate condition of Experiment 2 and Experiment 3 (see **Table 7**).

Because these conditions have the same number of consistent statements (4 sentences), what is compared are the type of statements in the two remaining sentences (filler vs.

**Table 6. Results of the Bonferroni-corrected pairwise comparisons between the three conditions of Experiment 3.**

| Comparisons | Mean difference | 95% Confidence Interval for Difference [Lower, Upper] | p value (Bonferroni-corrected) | $BF_{10,U}^{*}$ |
|---|---|---|---|---|
| High-Consistency vs. Intermediate-Consistency | 22.87 | [17.57, 28.17] | < .001 | 1.40e+11 |
| Intermediate-Consistency vs. Low-Consistency | 29.94 | [21.64, 38.23] | < .001 | 6.68e+8 |
| High-Consistency vs. Low-Consistency | 52.80 | [46.35, 59.26] | < .001 | 1.43e+21 |

*Note.* $BF_{10,U}^{*}$ = Uncorrected Bayes Factor

contradictory). We found that congruency scores in the intermediate condition of Experiment 2 ($M$ = 79.47, $SD$ = 9.10), in which the remaining two sentences served as space fillers are significantly higher than congruency scores in the intermediate condition of Experiment 3 ($M$ = 60.72, $SD$ = 15.76), in which the remaining two sentences served as contradictory inconsistent statements, $t(71.69)$ = 7.01, $p$ < .0005. Bayesian analysis showed that the data was more likely to be observed under the alternative hypotheses relative to the null hypotheses, $BF_{10}$ = 2.32e+7, suggesting the importance of the consistency of information. In other words, having inconsistent information makes it more difficult to reason about mental states than having a lower amount of ToM-relevant information (i.e., Experiment 2: having 4 pieces of mentalizing-relevant information versus. Experiment 3: having 6 pieces of mentalizing-relevant information).

## General discussion

### Main findings

In Experiment 1, we demonstrated that when making mental state attributions, the combined amount and consistency of information from which we base mental state inferences are important. This provided some insights into the specific mechanism by which ToM operates. Consistent with an information-theoretic approach, reducing uncertainty about an agent's mental states yielded higher congruency scores in participants' ToM performance. However, because Experiment 1 manipulated both the amount and consistency of information available simultaneously, it was unclear whether uncertainty was reduced when the amount of relevant information increased, the consistency of relevant information increased, or both.

### Amount of information

In Experiment 2, we manipulated the amount of relevant information that participants had access to. We found that only the low-quantity condition, in which information quantity was impoverished, significantly differed from the other two conditions. These findings indicate that when informational amount is moderate, participants are able to make relatively congruent mental state inferences. Increasing informational amount beyond this does little to significantly influence performance. Both the Bayesian and frequentist statistics demonstrate that high uncertainty in the form of impoverished *amount* of information significantly influences the ability to make inferences on another person's mental state. Interestingly, even with two consistent statements (i.e., low-quantity condition), the congruency scores were significantly higher than chance level (i.e., 50%), $t(46)$ = 14.92, $p$ < .001, Cohen's $d$ = 2.18, implying that a relatively small amount of relevant information can be enough to reason about others' mental states.

**Table 7. Snapshot summary of all conditions across all three experiments.**

| Experiment | Condition | Sentences |
|:---:|---|---|
| 1 | Low-Uncertainty | 6 Consistent |
| | Intermediate-Uncertainty | 2 Consistent + 4 Filler |
| | High-Uncertainty | 3 Consistent + 3 Inconsistent |
| 2 | High-Quantity | 6 Consistent |
| | Intermediate-Quantity | 4 Consistent + 2 Filler |
| | Low-Quantity | 2 Consistent + 4 Filler |
| 3 | High-Consistency | 6 Consistent |
| | Intermediate-Consistency | 4 Consistent + 2 Inconsistent |
| | Low-Consistency | 2 Consistent + 4 Inconsistent |

## Consistency of information

In Experiment 3, we manipulated only the consistency of information that participants had access to when making mental state inferences. The findings indicate that the consistency of information is a critical factor that informs the cognitive processes involved in mental state reasoning. Consistent with the findings of Experiment 1, both the Bayesian and frequentist statistics support the notion that manipulating informational consistency directly altered performance in congruency scores. In the high-consistency condition, when all the information provided was consistent with respect to an agent holding a particular set of mental states, congruency scores in attributing mental states were highest relative to the other conditions. When statements were most inconsistent with one other, as in the low-consistency condition, congruency scores were significantly lower. All three conditions varied from one another thus indicating that consistency of information impacts mental state reasoning. These findings provide us with a better understanding of *how* ToM may operate when exposed to impoverished or contradictory social stimuli.

While the findings from all three Experiments show the same pattern, some important differences are noteworthy. The findings of Experiment 1 (in which both information amount and information consistency were altered) are strong, but after teasing out the potential effects of the amount and consistency of information in Experiment 2 (amount) and Experiment 3 (consistency), an interesting set of results emerge. Namely, the effect observed in Experiment 2 is not as large as that in Experiment 1, suggesting that the inclusion of the consistency manipulation is influencing mental state attributions. And furthermore, comparing Experiment 2 with Experiment 3, it is once again clear that the result is showing the same pattern, but the effect is much larger in Experiment 3 in which only the consistency of information was manipulated. Such patterns imply the significance of consistency when reasoning about others' mental states under uncertainty.

## Information theory and ToM

Information theory focuses on quantifying information based on measuring the probability of events, and one of the key points is uncertainty [20]. Indeed, theory from nearly four decades of research suggests that people have a strong motivation to reduce uncertainty about mental states, especially in social contexts. Moreover, the quantity and nature of information to reduce this uncertainty is dynamic [21]. In any given social situation, there are a series of possible representations of another agent's mental state, each with varying levels of certainty. Because inferring another person's mental states requires determining the least ambiguous mental states from countless possible combinations of abstract mental states, it is inevitable that

mental state reasoning involves uncertainty. The response option with the highest and strongest level of certainty is ultimately chosen as representing the agents' mental states (for a model of ToM that supports this assumption, see [14]). Such uncertainty levels stem from the richness of the information provided, such that informational richness gradually reduces participants' uncertainty regarding an agents' mental states.

Thus, in theory, manipulating informational richness regarding another agent's thoughts, beliefs, desires, and intentions—should directly influence participants' mental state attributions. Such manipulations may include changing the amount or consistency of the information that may serve as cues to the agent's mental states. Humans have a remarkable ability to put a variety of observable behaviors together and seamlessly generate inferences about the unobservable thoughts, beliefs, intentions, and desires of others. Despite the scarcity of information that we often have access to, we generate accurate mental state inferences on a regular basis. The current findings seem to support an information theoretic perspective. The three studies in this paper clearly demonstrate that the way in which we make mental state attributions is by extracting qualitatively rich information from an otherwise impoverished and scarce informational source. That is, although the amount of information that we have access to is important, the consistency of that information appears to be the key element that allows us to generate mental state inferences from people's behaviours. Recall that in Experiment 2, only the amount of information was altered. Here, participants did need some information to make mental-state attributions (the intermediate-quantity condition helped increases in congruency scores from the low-quantity impoverished condition). But beyond this, in the high-quantity (informationally enriched) condition, participants did not have significantly higher congruency scores than in the intermediate-quantity condition.

The results of Experiment 3 (in which informational consistency was altered) show more drastic and consistent patterns with significant differences in congruency scores as the informational source becomes more enriched with consistent information. But as a means of comparison, consider the intermediate conditions of both Experiments 2 (i.e., intermediate-quantity) and 3 (i.e., intermediate-consistency). In both conditions, participants have the same number of consistent sentences with the difference being the remaining sentences (serving as either filler or contradictory sentences). Importantly, even though the amount of information relevant to mental state reasoning in Experiment 3 (i.e., 6 mental-state reasoning-related sentences) is larger than that in Experiment 2 (i.e., 4 mental-state reasoning-related sentences), the results of the exploratory analysis comparing the intermediate condition of Experiment 2 with the intermediate condition of Experiment 3 indicate that that the contradictory sentences (relative to the filler sentences) are more potent at reducing performance in congruency scores. What this comparison suggests is the critical importance of the consistency of information when making mental state inferences.

## Limitations and future directions

Despite contributing to an understanding of the cognitive processes involved in mental state attribution, the three studies presented here have several limitations. One potential limitation is that it is uncertain whether the experiments are definitively tapping into ToM per se; however, given the fact that follow up questions all require mental state inferences pertaining to the beliefs, thoughts, and feelings of another person, we are relatively confident that we are, indeed, capturing ToM processing. Additionally, there is an underlying assumption that the manipulations in all three experiments are directly exerting an effect on participant's uncertainty levels, but other measures will be necessary to confirm this assumption. When analyzing participants' reaction times, as an indirect measure of uncertainty, the results showed that

reaction times significantly differed by condition in the three experiments (see S3 Appendix for the reaction time analyses). Therefore, these results suggest that the manipulations influenced participants' uncertainty levels when making mental-state inferences. However, more direct measures, for example, having follow-up questions with a certainty index immediately after participants viewed each vignette, could provide converging evidence for how uncertainty factors into mental state computation.

Furthermore, it is unclear whether participants are making inferences about mental "states" or "traits". Given that participants are introduced to a character with "mental states" over several points in time, it remains unclear whether attributions about the individual are "state-level" attributions or "trait-level" attributions. While this is an important consideration, ToM abilities themselves are thought to influence conceptualizations of traits [22]. Indeed, mental states are often related to dispositional traits [23] and likely underly cognitive mechanisms that facilitate the understanding that traits can be causally linked with desires [24]. Nevertheless, follow-up questions that index whether participants believe the statement is true of the individual in that particular moment (state) versus an aspect of the individual's character (trait) could help clarify the distinction.

One further limitation of this study is that in the development of the vignettes, some sentences may provide a richer source of information about underlying mental states than others. To help mitigate this issue, a randomization technique for creating the vignettes was used. A random number generator was used to determine all aspects of the vignette (i.e., the gender and name of the character in the vignette, and whether or not the objects of the mental state inferences are a person, place, or thing; see S2 Appendix). Critically, for conditions in which some sentences are consistent and others are either filler or inconsistent sentences, the consistent sentences were selected randomly from the total list. Therefore, this process ensures that if there is a sentence that is particularly salient and informationally rich, it has the same likelihood of appearing in all experimental conditions as other sentences. This way, the potential that some sentences may resonate more with a participant than others is controlled for through randomization.

Importantly, the stimuli and sentences themselves are wanting of a manipulation check to ensure that they are, in fact, consistent (or inconsistent) and if so, how much each sentence contributes to participants' mental state inferences. A follow-up study that further investigates whether and how participants weigh the available pieces of information will be necessary to clearly address the limitation. For example, using a rating design in which asking participants to quantify how much each single piece of information is consistent will elaborate the current findings. In this way, one could quantify, for each single trial, how much and how consistent the available information is.

In these experiments, we have used information theory accounts to motivate our research questions and hypotheses. To directly link to information theory, manipulating diverse features of information, quantifying them, and running a computational modeling algorithm would be beneficial. For example, features of information related to uncertainty (e.g., the amount of different types of information, the reliability of information source, and the contexts) can be manipulated and quantified for computational modeling, thereby giving more direct evidence supporting an information theoretic perspective as a cognitive mechanism of ToM.

One potential avenue for future research is to determine whether the same factors are at play in children with a potentially underdeveloped ability to make accurate mental state attributions. Future research can also explore the extent to which the order in which various statements appear influence mental state attributions. Another interesting and clinically relevant avenue for future research is to investigate whether uncertainty-reduction operates similarly in

individuals with an autism diagnosis, who make mental state inferences in different ways than healthy controls. Furthermore, intervention research can use these insights to train those who struggle with ToM problems by focusing on techniques to extract highly consistent, rich pieces of information from a given stimulus and thereby generate accurate attributions about the mental states of others.

Despite the limitations outlined in the stimuli used within the three experiments of the current study, there are several features of the stimuli that are noteworthy. Other existing ToM tasks involve one definitive correct answer to a single ToM question (e.g., Sally-Anne task in which participants are asked to infer another agent's mental state on an object's location; location A or B?). In contrast, the type of ToM tasks used in the current study allows for a plethora of opportunities to infer mental states with multiple questions targeting various mental state processes. This process provided us with a richer response for our primary outcome (i.e., rather than "location A/B" in previous ToM tasks) and is more reflective of mental state reasoning in every day, real life processes. That is, we typically do not simply ask ourselves a binary question pertaining to another individual's mental states. But rather, we integrate the complexities of people's mental states with a series of inferences on what the other person thinks, how they feel, what they believe to be true, and how they will likely behave in the future. We are also often faced with competing and complex information that may alter our mental state inferences, features which our task and manipulations capture. Furthermore, with our task, there can always be individual differences in which statement are perceived as more rich when inferring others' mental states. To address this in the current set of studies and rule out any individual difference factors, we randomized many aspects of the stimuli including which sentences were chosen to appear in the vignettes of the other conditions. Therefore, it is unlikely that one informationally rich statement drove any effects observed, as the likelihood of each sentence being chosen in other conditions was random, as was the likelihood of a participant being exposed to each experimental condition.

## Conclusion

Across all three Experiments, performance on measures of mental state reasoning changed as a function of our manipulations (to varying degrees). Uncertainty-reduction is the most plausible mechanism by which this occurs. From an information-theoretic perspective, the cognitive processes underlying ToM generate a series of possible representations regarding another agent's mental states. Each of these representations have an attached level of certainty [14, 25]. Because mental states are not visible, they must be inferred. Therefore, the representation with the strongest level of certainty is chosen as correctly representing another agent's mental states. In effect, by extracting the most relevant and qualitatively rich pieces of information from the behaviour of others—we reduce our uncertainty pertaining to their mental states and can thereby influence mental state attributions. The current findings, as an exploratory study, found that both the amount and consistency of information seem to influence our mental state attributions, with the consistency of information being a more salient feature when inferring mental states of others.

## Supporting information

**S1 Appendix. Examples of vignette scripts.** *Notes*. Participants read 15 vignettes, each with 3 conditions totaling to 45 vignettes depicted here; participants only viewed one condition of each vignette (i.e., 15 vignettes). The following is examples of 45 vignette scripts used in Experiment 1. For Experiments 2 and 3, the same vignettes were used with alterations in the number

of filler, consistent, or inconsistent statements.
(DOCX)

**S2 Appendix. Randomization technique for vignettes.**
(DOCX)

**S3 Appendix. Reaction time data analyses.**
(DOCX)

## Acknowledgments

We thank all the participants who participated in the research, the research assistants who helped with data collection, and Dr. Adam S. Cohen who provided valuable support and insights contributing to the research (i.e., conceptualization, study design, ethics approval institution, and comments on an earlier draft).

## Author Contributions

**Conceptualization:** Isu Cho, Nellie Kamkar, Niki Hosseini-Kamkar.

**Data curation:** Isu Cho, Nellie Kamkar.

**Formal analysis:** Isu Cho, Nellie Kamkar.

**Investigation:** Isu Cho, Nellie Kamkar, Niki Hosseini-Kamkar.

**Methodology:** Isu Cho, Nellie Kamkar, Niki Hosseini-Kamkar.

**Project administration:** Isu Cho, Nellie Kamkar.

**Validation:** Niki Hosseini-Kamkar.

**Writing – original draft:** Nellie Kamkar.

**Writing – review & editing:** Isu Cho, Nellie Kamkar, Niki Hosseini-Kamkar.

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
