## [Decision Letter · Decision Letter 0]

25 Jul 2022

PONE-D-22-13954Reasoning About Mental States Under UncertaintyPLOS ONE

Dear Dr. Kamkar,

Thank you for submitting your manuscript to PLOS ONE. After careful consideration, we feel that it has merit but does not fully meet PLOS ONE’s publication criteria as it currently stands. Therefore, we invite you to submit a revised version of the manuscript that addresses the points raised during the review process.

 All three expert revisions have found positive aspects in your manuscript. However, they also raise several important issues, many of which seem easily manageable, but some of which address both the theory and methodology. Please make sure to carefully address all of the reviewers' points before resubmitting your manuscript.

We look forward to receiving your revised manuscript.

Kind regards,

Enrico Toffalini, Ph.D

Academic Editor

PLOS ONE

Journal Requirements:

2. PLOS requires an ORCID iD for the corresponding author in Editorial Manager on papers submitted after December 6th, 2016. Please ensure that you have an ORCID iD and that it is validated in Editorial Manager. To do this, go to ‘Update my Information’ (in the upper left-hand corner of the main menu), and click on the Fetch/Validate link next to the ORCID field. This will take you to the ORCID site and allow you to create a new iD or authenticate a pre-existing iD in Editorial Manager. Please see the following video for instructions on linking an ORCID iD to your Editorial Manager account: https://www.youtube.com/watch?v=_xcclfuvtxQ.

4. Please provide additional details regarding ethical approval and participant consent in the body of your manuscript. In the Methods section, please ensure that you have specified:

(1) the name of the IRB/ethics committee that approved your study;

and for each experiment (2) whether consent was informed and (3) what type of consent you obtained (for instance, written or verbal).

Reviewers' comments:

Reviewer's Responses to Questions

**Comments to the Author**

1. Is the manuscript technically sound, and do the data support the conclusions?

Reviewer #1: Yes

Reviewer #2: Partly

Reviewer #3: Partly

2. Has the statistical analysis been performed appropriately and rigorously? 

Reviewer #1: Yes

Reviewer #2: Yes

Reviewer #3: Yes

3. Have the authors made all data underlying the findings in their manuscript fully available?

Reviewer #1: Yes

Reviewer #2: Yes

Reviewer #3: Yes

4. Is the manuscript presented in an intelligible fashion and written in standard English?

Reviewer #1: Yes

Reviewer #2: Yes

Reviewer #3: Yes

5. Review Comments to the Author

Reviewer #1: The paper describes three experiments that suggest that both quantity and quality of information are important features of social stimuli, but that the quality of available information plays a greater role in inferring the mental state of others. The study seems interesting to me, but it is not clearly described and I am not in a position to fully evaluate it at this time. I hope that my suggestions will help the authors to clarify the experimental aspects of their research.

Please indicate what your hypotheses are at the end of the introduction or at the beginning of the description of each study.

Study 1

Please include a brief introduction to Study 1 (as you did for Studies 2 and 3) to explain what you investigated in this particular study.

For all studies

Please explain in the experimental material what relates to the quantity and what relates to the quality of information that is critical for changing mental state attributions.

It is not clear to me how the authors were able to investigate "whether the quantity (i.e., amount) of information alone, the quality (i.e., type) of information alone, or both are crucial for changing mental state attributions" since the variable manipulated in the experiment appears to be the degree of uncertainty. Please use the labels in a coherent way to indicate what you want to study and what you studied in the experiment.

Please include a section on statistical analysis to clarify how you statistically examined your hypotheses.

In the introduction (and not just before each experiment), please make it clear what specific hypothesis you wanted to test for each experiment and what variable you manipulated in each experiment. Perhaps it would help to schematize these aspects in a table. Also, it might be helpful to the reader if you indicate in each table what is being investigated.

Why do the authors refer to quantity and quality of information in the general introduction and manipulate the variable consistency of information in Experiment 3? Is consistency of information part of the quality or quantity of information? I suspect quality, but the authors need to be clear from the beginning which variable was manipulated in the three experiments and explain in the introduction what they wanted to study in each experiment, how they studied it, and what hypothesis was tested in each of them.

Reviewer #2: In this paper, the authors conduct three experiments designed to test the impact of the amount and quality (i.e., consistency) of information when making mental state inferences. They find that reducing the amount, and separately, the quality of the information both reduces the accuracy of mental state inferences with the impact being greater for quality than amount.

The studies address a fascinating and understudied set of questions in the theory of mind literature. It brings to mind other fruitful avenues of inquiry on theory of mind such as predictive coding accounts of mental state inferences. There’s also much to admire in the manuscript itself. The paper is clearly written and I appreciate the brevity. I also commend the authors for their preregistration and open science practices.

I have one primary concern about the studies. The “correct” answers are determined on the basis of those scenarios in which all the information is provided/is consistent. I’m not sure if that’s a valid way of saying what’s correct or incorrect in these scenarios, particularly in the case of the consistency manipulation. For example, with several bits of “inconsistent" information, it might turn out to be that what is considered a wrong answer is actually most accurate. That is, if given 4 bits of information considered “inconsistent" and 2 bits considered “consistent," then rationally, the accurate answer regarding the character’s mental state would be in line with the “inconsistent” information. More generally, there’s no ground truth to the mental states experienced by the characters. In fact, what might be classified as “inconsistent” could reasonably be interpreted as consistent or vice-versa (e.g., Lewis gives Clara the advertisement about the local orchestra not because he think she’s great, but because he doesn’t want her playing in the house and thinks that by playing in the local orchestra, she’ll be less likely to practice at home and/or get access to the orchestra’s practice facilities). Moreover, it might be that people tend to weight certain bits of information more heavily than other when making mental state inferences (e.g., statements that speak to internal sensations - using noise cancelling headphones - versus more ambiguous behaviors - paying to have a piano professionally tuned). All of these possibilities make me wonder about what accuracy, as currently assessed, actually tells us, and whether the task should be scored differently. For example, accuracy could be determined based on general consensus by having a large sample state which responses are accurate given the information.

Reviewer #3: The authors examine how adults use different piece of information to infer mental states. I find the idea of studying how people incorporate all sorts of information into their mental state reasoning fascinating because I agree with the authors that researchers often narrow their focus to one specific cause of a mental state so we know little about how people actually use multiple piece of (sometimes conflicting) information. Despite my excitement about this idea, my overall evaluation of the paper is less positive. In particular, I feel that the scope of the paper is less than promised by the Introduction. I also have some theoretical and methodological concerns.

Theoretically, I think it will be important for the authors to consider the trait attribution literature. Some discussion of why we would expect people to use inconsistent versus consistent information to be the same versus different across mental states and traits is needed. This feels especially necessary because the participants are actually introduced to a person with “mental states” over several points in time which to me feels more trait-like (i.e., what is this person’s stable attitudes?) than mental-state like (i.e., what is this person feeling in the here-and-now?). As it stands, I am not clear what we have learned about how people process consistent and inconsistent information about a person over time.

The authors appear especially interested in how uncertainty shapes people’s ability to engage in mental state reasoning. I think many of their claims about “uncertainty” are dependent on whether the participants actually felt uncertain. I would recommend that the researchers run similar studies where they actually measure whether participants felt more or less certain across conditions. The authors acknowledge this in their Discussion, but given that they are interested in uncertainty, I think data is needed to speak to this concern. Alternatively, I see that the authors have reaction time data, they could potentially use this to speak to uncertainty.

I would recommend not using the term “accuracy” but rather just saying that the date were coded in line with the idea that Lewis likes Clara’s piano playing. To me, I find it confusing to say that the “high-uncertainty” condition is less accurate than the “low-uncertainty” condition, it more seems like they have different interpretations because they have different information. I certainly wouldn’t want to say that the participants in the “high-uncertainty” condition were worse mental-state reasoners than those in the “low-uncertainty” condition.

The authors use many different words throughout to (I think) mean the same thing (consistency, type, quality), this makes the arguments and conditions of the paper a little difficult to follow.

I was surprised to see that the authors randomized the order that the different information came in given that prior research suggests that the order people learn information matters (e.g., Cone, Flaharty, & Ferguson, 2021). Is it possible for the authors to explore differences in people’s mental state attributions by order of information?

I am a little confused about a couple of aspects of the design. First, when the authors say that the order of conditions was counterbalanced, does this mean that they were blocked (so, for example, all of the intermediate, then all of the low-uncertainty, and then all of the high-uncertainty) or were all of the stories counterbalanced so that participants might get an intermediate story, and then a low-uncertainty, and then another intermediate story, and then a high-uncertainty, etc.? If the stories were blocked, please include an order effect analysis. This would be especially important if the authors want to use the reaction time data as evidence of uncertainty because it's possible that people's certainty went up or down based on what condition they saw first.

I also wasn’t sure if the 6 different “consistent” statements from the “low-uncertainty” condition could be randomly slotted into each of the “consistent” statements for the intermediate and high-uncertainty conditions or whether which “consistent” statements were used in the intermediate and high-uncertainty conditions were fixed across participants. This seems important to know especially because it does not appear that the authors have pilot tested these statements to ensure that they all provide similar information.

It would be helpful if the authors could provide all of the vignettes that were used. Was it the case that the “correct” mental state was always positive (e.g., Lewis like’s Clara’s piano playing) or was it sometimes the case that the person would have a negative mental state towards the person? Relatedly, were the characters’ mental states always social (i.e., what they thought of another person) or were they sometimes asocial (e.g., their belief about where an object is). These are important pieces of information to report in the manuscript.

The authors argue that the findings from Experiment 1 and 2 are similar. I’m not sure I agree with this. Yes, they get the same pattern, but the effect seems much larger in Experiment 1. This should be addressed.

6. PLOS authors have the option to publish the peer review history of their article (what does this mean?). If published, this will include your full peer review and any attached files.

Reviewer #1: No

Reviewer #2: No

Reviewer #3: No

---

## [Author Response · Author response to Decision Letter 0]

30 Aug 2022

EDITOR'S COMMENTS

Comment 1.

Response 1. We would like to thank the editor for bringing this to our attention. We have revised our manuscript in accordance with PLOS ONE’s style requirements, including file naming conventions required. We have also updated all of our references in accordance with PLOS ONE’s referencing guidelines. 

Comment 2.

PLOS requires an ORCID iD for the corresponding author in Editorial Manager on papers submitted after December 6th, 2016. Please ensure that you have an ORCID iD and that it is validated in the Editorial Manager. To do this, go to ‘Update my Information’ (in the upper left-hand corner of the main menu), and click on the Fetch/Validate link next to the ORCID field. This will take you to the ORCID site and allow you to create a new iD or authenticate a pre-existing iD in Editorial Manager. Please see the following video for instructions on linking an ORCID iD to your Editorial Manager account: https://www.youtube.com/watch?v=_xcclfuvtxQ.

Response 2. We have now included an ORCID iD for the corresponding author (ID 0000-0001-6001-8745) and validated it in the Editorial Manager. 

Comment 3.

Please include captions for your Supporting Information files at the end of your manuscript, and update any in-text citations to match accordingly. Please see our Supporting Information guidelines for more information: http://journals.plos.org/plosone/s/supporting-information.

Response 3. We would like to thank the editor for this comment and have addressed it by including captions for Supporting Information files at the end of our revised manuscript in accordance with the guidelines provided. 

Comment 4.

Please provide additional details regarding ethical approval and participant consent in the body of your manuscript. In the Methods section, please ensure that you have specified: (1) the name of the IRB/ethics committee that approved your study; and for each experiment (2) whether consent was informed and (3) what type of consent you obtained (for instance, written or verbal).

Response 4. We agree completely with the editor and have since provided the following statement in the Methods section of each experiment:

Institutional Review Board (IRB) ethical approval was obtained by Western University’s Research Ethics Board (REB) for non-medical research (file # 104311). All participants provided written informed consent prior to participating in the study.

[Page 6, Line 127-129].

Comment 5. Please include your full ethics statement in the ‘Methods’ section of your manuscript file. In your statement, please include the full name of the IRB or ethics committee who approved or waived your study, as well as whether or not you obtained informed written or verbal consent. If consent was waived for your study, please include this information in your statement as well.

Response 5. As with the previous comment, we agree and would like to thank the editor for this comment. Like Experiment 1, in the Methods sections of Experiments 2 and 3, we have added the following sentence:

IRB ethical approval was obtained by Western University’s REB for non-medical research (file # 104311). All participants provided written informed consent prior to participating in the study.

[Page 16, Line 290-292].

[Page 22, Line 381-383].

REVIEWERS' RESPONSES

Reviewer's Responses to Questions & Comments to the Author

1. Is the manuscript technically sound, and do the data support the conclusions?

Reviewer #1: Yes

Reviewer #2: Partly

Reviewer #3: Partly

2. Has the statistical analysis been performed appropriately and rigorously?

Reviewer #1: Yes

Reviewer #2: Yes

Reviewer #3: Yes

3. Have the authors made all data underlying the findings in their manuscript fully available?

Reviewer #1: Yes

Reviewer #2: Yes

Reviewer #3: Yes

4. Is the manuscript presented in an intelligible fashion and written in standard English?

Reviewer #1: Yes

Reviewer #2: Yes

Reviewer #3: Yes

REVIEWER #1's COMMENTS

Comment 1.

The paper describes three experiments that suggest that both quantity and quality of information are important features of social stimuli, but that the quality of available information plays a greater role in inferring the mental state of others. The study seems interesting to me, but it is not clearly described and I am not in a position to fully evaluate it at this time. I hope that my suggestions will help the authors to clarify the experimental aspects of their research.

Please indicate what your hypotheses are at the end of the introduction or at the beginning of the description of each study.

Response 1. We would like to thank the reviewer for this constructive feedback. We have now addressed this comment by including a section on study questions and hypotheses for each experiment, and ensuring that existing sections include details of predictions and independent and dependent variables, as follows [added sections are in darker font here for illustrative purposes]

Experiment 1

In this experiment, the independent variable was the degree of uncertainty with three levels—low, intermediate, and high uncertainty. Critically, uncertainty was manipulated using both the amount and consistency of information. The dependent variable was accuracy on questions regarding an agent’s mental state. We predict a direct effect of uncertainty manipulation on ToM performance such that as one’s uncertainty regarding an agent’s mental states decreases, their accuracy on a test of ToM will also increase; whereas, as one’s uncertainty regarding an agent’s mental state increases, the accuracy will decrease. 

 [Page 6, Line 117-123].

Experiment 2

In Experiment 2, we aimed to discern whether specifically and selectively altering only the amount of relevant information would alter participants’ mental state attributions. We therefore altered participants’ certainty regarding an agent’s mental states by manipulating the quantity of information provided to them only—thus the independent variable was informational quantity with three levels, low, intermediate, and high. We then measured participants’ ability to reason about an agents’ mental states as a function of varying the amount of relevant information provided in the stimulus. From an information-theoretic perspective, it was predicted that as more relevant behavioral information is provided to support a mental state inference, accuracy of mental state reasoning would increase.

 [Page 15, Line 281-282].

Experiment 3

In Experiment 3, we aimed to discern whether specifically and selectively altering only

the consistency of relevant information would alter participants’ mental state attributions.

We therefore measured participants’ ability to reason about an agents’ mental states by

manipulating the consistency of information provided to them only. Here, the

independent variable was consistency of information, with three levels, low, intermediate,

and high. From an information-theoretic perspective, it was predicted that as the

behavioral information provided becomes consistent with supporting a mental state

inference, accuracy of mental state reasoning would increase.

 [Page 21, Line 367-368]. 

Comment 2.

Study 1, please include a brief introduction to Study 1 (as you did for Studies 2 and 3) to explain what you investigated in this particular study.

Response 2. We have included this section as per the reviewer’s helpful suggestion. Specifically, in addition to the General Introduction of the revised manuscript, there now appears a section under Experiment 1 with a brief introduction of the first experiment, as follows:

 Experiment 1

In order to make a mental state inference about the agent in the vignette, participants require information that is consistent with the set of mental state inferences determined at the outset of the trial (i.e., that Lewis likes Clara's piano playing). In Experiment 1, the number of relevant sentences was manipulated by inserting filler sentences that have nothing to do with the agent's mental states (i.e., that Lewis speaks four languages). In addition, the consistency of sentences was manipulated by inserting sentences that are directly opposite to the mental state inferences of the agent (i.e., Lewis wears noise-canceling headphones when Clara plays the piano). In this experiment, the independent variable was the degree of uncertainty with three levels—low, intermediate, and high uncertainty. Critically, uncertainty was manipulated using both the amount and consistency of information. The dependent variable was accuracy on questions regarding an agent’s mental state. We predict a direct effect of uncertainty manipulation on ToM performance such that as one’s uncertainty regarding an agent’s mental states decreases, their accuracy on a test of ToM will also increase; whereas, as one’s uncertainty regarding an agent’s mental state increases, accuracy will decrease.

[Pages 5-6, Line 157-179].

Comment 3.

For all studies, please explain in the experimental material what relates to the quantity and what relates to the quality of information that is critical for changing mental state attributions.

Response 3. This is an excellent question. The quality entails the consistency of the information provided, whereas the quantity pertains to the number of sentences. To manipulate the consistency of information provided, sentences that are directly opposite to the predetermined mental states are inserted, whereas to manipulate the amount of information provided, filler (irrelevant) sentences are inserted. What we are manipulating is the degree of uncertainty in Experiment 1 and in Experiments 2 and 3, we aim to discern where the uncertainty comes from (the consistency of information, or the number of statements). We have addressed this comment by adding the following statements to each Experiment’s Introductory section to add further clarity on what is being manipulated [added sections are in darker font here for illustrative purposes]

Experiment 1

In order to make a mental state inference about the agent in the vignette, participants require information that is consistent with the set of mental state inferences determined at the outset of the trial (i.e., that Lewis likes Clara's piano playing). In Experiment 1, the number of relevant sentences was manipulated by inserting filler sentences that have nothing to do with the agent's mental states (i.e., that Lewis speaks four languages). In addition, the consistency of sentences was manipulated by inserting sentences that are directly opposite to the mental state inferences of the agent (i.e., Lewis wears noise-canceling headphones when Clara plays the piano). 

[Page 5-6, Line 157-173].

Experiment 2

Like Experiment 1, for participants to make a mental state inference about the agent in the vignette, they require information that is consistent with the set of mental state inferences determined at the outset of the trial (i.e., that Lewis likes Clara's piano playing). In Experiment 2, only the number of relevant sentences was manipulated by inserting filler sentences that have nothing to do with the agent's mental states (i.e., that Lewis speaks four languages).

[Page 15, Line 361-365].

Experiment 3

Like Experiments 1 and 2, to make mental state inferences that are in line with those pre-set at the outset of the trial, participants require information that is consistent (i.e., that Lewis likes Clara's piano playing). In Experiment 3, only the consistency of sentences was manipulated by inserting sentences that are directly opposite to the pre-set mental state inference (i.e., that Lewis wears noise-canceling headphones when Clara plays the piano). 

 [Page 21, Line 504-508].

Comment 4.

It is not clear to me how the authors were able to investigate "whether the quantity (i.e., amount) of information alone, the quality (i.e., type) of information alone, or both are crucial for changing mental state attributions" since the variable manipulated in the experiment appears to be the degree of uncertainty. Please use the labels in a coherent way to indicate what you want to study and what you studied in the experiment.

Response 4. We agree with this reviewer that further clarity in the labels and wording is warranted, and thus, we have revised the wording of all 3 Experiments. This relabeling is reflected throughout the revised manuscript, as well as in Table 7, as follows:

(1) For Experiment 1, we have relabeled the 3 conditions to highlight that the degree of uncertainty is the variable being manipulated (as this reviewer rightly pointed out).

For Experiment 1, there are 3 conditions: 

Low-Uncertainty

Intermediate-Uncertainty

High-Uncertainty

(2) For Experiment 2, we have relabeled the 3 conditions to highlight that the degree of uncertainty is being manipulated via alterations only in the quantity of information. In Experiment 2, there are 3 conditions: 

High-Quantity

Intermediate-Quantity

Low-Quantity

(3) For Experiment 3, we have relabeled the 3 conditions to highlight that the degree of uncertainty is being manipulated via alterations only in the consistency of information. In Experiment 3, there are 3 conditions: 

High-Consistency

Intermediate-Consistency

Low-Consistency.

Comment 5.

Please include a section on statistical analysis to clarify how you statistically examined your hypotheses.

Response 5. We have addressed this comment by including a section on statistical analyses conducted to examine each hypothesis as follows [added sections are in darker font here for illustrative purposes].

In the Design section of Experiment 1:

Experiment 1 was conducted using a within-subjects experimental design with 3 conditions. There was one within-subject independent variable, degree of uncertainty, with three levels: low, intermediate, and high. To address the hypothesis of a direct effect of the uncertainty manipulation on responses to questions pertaining to the agent's mental states, a repeated-measures ANOVA is required (with follow-up pairwise tests if applicable). In addition to frequentist statistics, Bayesian statistical analyses (Bayes factor) will be conducted to determine differences in participants' responses to the mental state inference questions across the three conditions.

[Page 7, Line 199-206].

In the Design section of Experiment 2:

The design in Experiment 2 was similar to Experiment 1 with the following modifications. Experiment 2 was conducted using a within-subjects experimental design with 3 conditions. In the intermediate-quantity condition, four of the six total sentences provided relevant information about the mental states of an agent. The remaining two sentences were irrelevant filler sentences. In the high-quantity condition, all six sentences that participants were provided with were consistent with mental state(s). This condition provided participants with the highest amount of information to support mental state inferences. In the low-quantity condition, only two of the six total sentences were consistent with the character’s “correct” set of mental states (i.e., the mental states corresponding to the consistent information), with the majority of sentences (the remaining four) consisting of irrelevant filler sentences. To address the hypothesis of a direct effect of manipulating uncertainty via direct alterations of the quantity of consistent sentences, a repeated-measures ANOVA is required (with follow-up pairwise tests if applicable). In addition to frequentist statistics, Bayesian statistical analyses (Bayes factor) will be conducted to determine differences in participant's responses to the mental state inference questions across the three conditions.

[Page 16, Line 397-414].

At the end of the Design section of Experiment 3:

 Like Experiments 1 and 2, Experiment 3 was conducted using a within-subjects experimental design with 3 conditions. To address the hypothesis of a direct effect of manipulating uncertainty via direct alterations of the consistency of statements (i.e., including inconsistent statements), a repeated-measures ANOVA is required (with follow-up pairwise tests if applicable). In addition to frequentist statistics, Bayesian statistical analyses (Bayes factor) will be conducted to determine differences in participant’s responses to the mental state inference questions across the three conditions. 

[Page 21, Line 515-522].

Comment 6.

In the introduction (and not just before each experiment), please make it clear what specific hypothesis you wanted to test for each experiment and what variable you manipulated in each experiment. Perhaps it would help to schematize these aspects in a table. Also, it might be helpful to the reader if you indicate in each table what is being investigated.

Response 6. We have addressed the reviewer’s comments by including a section in the general introduction of the paper that outlines our study questions, and our hypotheses. Within each subsection of each experiment, we have also explained what the independent variable manipulated and what our predictions were. We have also relabeled the tables in the manuscript to ensure that what is being investigated is clear to the reader. The following is the section of text addressing this comment in the general introduction of the revised manuscript [added sections are in darker font here for illustrative purposes].

Here, we use an information-theoretic perspective to better understand the mechanisms underlying adult mental state reasoning abilities. An information-theoretic perspective to understanding ToM asserts that ToM processing generates a series of possible representations of another agent’s mental state, each with varying levels of certainty [reference included]. To determine whether ToM operates through gradual uncertainty-reduction, we alter participants’ certainty regarding an agent’s mental state and measure their ability to reason about those mental states thereafter. In Experiment 1, participants’ certainty levels regarding an agent’s mental states were experimentally manipulated by altering the informational richness of the stimulus. Specifically, both the amount and type of information provided to participants were altered and participants’ responses to reason about the agent’s mental states were analyzed. We predict that altering one's uncertainty about an agent's mental states will have a direct impact on their performance on a test of ToM, with accuracy (please see the design section in Experiment 1 for detailed explanation on the definition of accuracy) increasing as one's uncertainty about the agent's mental state decreases and decreasing as one's uncertainty about the agent's mental state increases. In two follow-up experiments, we aimed to discern whether the quantity (i.e., amount) of information alone, the quality (i.e., consistency) of information alone, or both are critical in altering mental state attributions. In Experiment 2, we aimed to determine whether specifically and selectively varying only the amount—or quantity—of information will influence ToM. We predict that manipulating the quantity of information will have a direct effect on ToM performance such that as the amount of information provided increases, their accuracy on ToM tests will also increase. In Experiment 3, we determined whether specifically and selectively altering the consistency of information contributed to performance on a ToM task. Here, consistency of information was manipulated by providing participants with varying degrees of statements that were either consistent or contradictory with the agent’s mental state. We predicted that the consistency of information would directly influence performance on ToM, such that receiving more consistent information would yield greater accuracy on the ToM tasks. Prior to collecting data, we preregistered our hypotheses, methods, and analysis plans for all three studies on the Open Science Framework (https://osf.io/89wr4/). Analyses conducted for all three studies involve a priori, hypothesis-driven analysis plans.

[Pages 4-5, Line 123-154].

Comment 7.

Why do the authors refer to quantity and quality of information in the general introduction and manipulate the variable consistency of information in Experiment 3? Is consistency of information part of the quality or quantity of information? I suspect quality, but the authors need to be clear from the beginning which variable was manipulated in the three experiments and explain in the introduction what they wanted to study in each experiment, how they studied it, and what hypothesis was tested in each of them.

Response 7. We would like to thank this reviewer for this perceptive comment. The reviewer is correct that the quality was referring to the consistency and these words were used interchangeably in the original version of the submitted manuscript. We have since revised the manuscript throughout to refer to the uncertainty manipulations as those altering the “amount” or the “consistency” of information and taken out the quality-quantity terminology altogether. We have also included a section at the beginning of each experiment delineating what was studied, how it was studied, and the hypotheses made for each experiment. The response to Reviewer 1’s Comment 1 includes these additions. They can also all be found in the tracked manuscript (the paragraphs appearing after each subheading of Experiment 1, Experiment 2, and Experiment 3).

REVIEWER #2's COMMENTS

Comment 1.

In this paper, the authors conduct three experiments designed to test the impact of the amount and quality (i.e., consistency) of information when making mental state inferences. They find that reducing the amount, and separately, the quality of the information both reduces the accuracy of mental state inferences with the impact being greater for quality than amount.

The studies address a fascinating and understudied set of questions in the theory of mind literature. It brings to mind other fruitful avenues of inquiry on theory of mind such as predictive coding accounts of mental state inferences. There’s also much to admire in the manuscript itself. The paper is clearly written and I appreciate the brevity. I also commend the authors for their preregistration and open science practices.

I have one primary concern about the studies. The “correct” answers are determined on the basis of those scenarios in which all the information is provided/is consistent. I’m not sure if that’s a valid way of saying what’s correct or incorrect in these scenarios, particularly in the case of the consistency manipulation. For example, with several bits of “inconsistent" information, it might turn out to be that what is considered a wrong answer is actually most accurate. That is, if given 4 bits of information considered “inconsistent" and 2 bits considered “consistent," then rationally, the accurate answer regarding the character’s mental state would be in line with the “inconsistent” information. More generally, there’s no ground truth to the mental states experienced by the characters. In fact, what might be classified as “inconsistent” could reasonably be interpreted as consistent or vice-versa (e.g., Lewis gives Clara the advertisement about the local orchestra not because he think she’s great, but because he doesn’t want her playing in the house and thinks that by playing in the local orchestra, she’ll be less likely to practice at home and/or get access to the orchestra’s practice facilities). Moreover, it might be that people tend to weight certain bits of information more heavily than other when making mental state inferences (e.g., statements that speak to internal sensations - using noise cancelling headphones - versus more ambiguous behaviors - paying to have a piano professionally tuned). All of these possibilities make me wonder about what accuracy, as currently assessed, actually tells us, and whether the task should be scored differently. For example, accuracy could be determined based on general consensus by having a large sample state which responses are accurate given the information.

Response 1. We would like to thank Reviewer 2 for the kind compliments on our work and for the enthusiastic comments on our research. The comments pertaining to the informational richness of each sentence were very interesting and thought provoking. This reviewer is absolutely correct that some sentences may be more likely to elicit a mental state inference than others (despite both being classified as “consistent”). To address this, we have added the following to the limitations section of the manuscript.

One further limitation of this study is that in the development of the vignettes, some sentences may provide a richer source of information about underlying mental states than others. To help mitigate this issue, a randomization technique for creating the vignettes was used. A random number generator was used to determine all aspects of the vignette (i.e., the gender and name of the character in the vignette, and whether or not the objects of the mental state inferences are a person, place, or thing; see S2 Appendix). Critically, for conditions in which some sentences are consistent and others are either filler or inconsistent sentences, the consistent sentences were selected randomly from the total list. Therefore, this process ensures that if there is a sentence that is particularly salient and informationally rich, it has the same likelihood of appearing in all experimental conditions as other sentences. This way, the potential that some sentences may resonate more with a participant than others is controlled for through randomization.

[Pages 33-34, Line 816-832].

We also found the comment pertaining to the interpretation of consistency very interesting. Specifically, the notion that Lewis may take the advertisement to the local orchestra home to Clara simply because he would like her to play the piano outside of their home was a reasonable interpretation. That said, this particular vignette was one of 15 that was alluded to for illustrative purposes in the text of the manuscript. Participants received an additional 14 vignettes that did not include this possibility. For example, one vignette describes a character who does not prefer the heat and participants are asked to determine whether or not she will plan a trip to a hot destination. We have provided all vignettes as an appendix in the supplemental information to provide further clarity for the reader.

And finally, we agree completely with the highly relevant and important comment pertaining to the term “accuracy”. This reviewer is correct that when the “inconsistent” pieces of information are provided, they actually sway the participant to respond in a way that harmonizes with the information given and thus, is not necessarily an inaccurate response. In the original manuscript, we predefined accurate responses as those corresponding to the answers when only consistent information was provided (please see the paragraph below on page 10) [sentences that are bolded are for emphasis in this letter]

To obtain an accuracy measure for each condition from participants, the five responses to the test questions (believe, think, know, emotion, and intention) on each trial were aggregated—given five trials per condition, this resulted in an accuracy score out of twenty-five. The “correct” set of mental state attributions were determined a priori regardless of the type of condition (i.e., low/high/intermediate uncertainty conditions). Correct answers were specified as the ones corresponding to the answers when only consistent information was provided. For example, at the outset of the trial, the experimenters pre-set "correctness" to mean that the character Lewis likes Clara's piano playing. Thus, lower accuracy in the high uncertainty condition indicates participants’ responses reflected the information inconsistent with the pre-set “correct” response. Our hypotheses pertained to the extent to which the amount and type of information that participants are given influences their final mental-state attributions.

To avoid any confusion in understanding the term “accuracy” and to reflect the reviewer’s comment, in addition to this explanation, we have now added the following description at the end of the Design section of Experiment 1 in which the task is described for all three Experiments. This description helps clarify the interpretation of the accuracy score for all three Experiments and we feel that it will significantly improve the readers’ understanding of participant responses. We would therefore like to express our gratitude to this reviewer for pointing this out.

 Therefore, it is noteworthy that the term "accuracy" is not necessarily reflective of true correct responses, as the experimenters' pre-set list of "correct" responses are by no means objectively true and accurate statements. That is, in conditions in which more "inconsistent" information is provided (i.e., that Lewis wears noise canceling headphones when Clara plays the piano), participants may respond in ways that indicate that Lewis does not like Clara's piano playing. While this response pattern is, in effect, correct, in the present set of Experiments the reference term of "accuracy" being low simply reflects the extent to which participants agree with the response options that have been predetermined a-priori. This understanding of the accuracy score is integral to interpreting the response options of participants in all three Experiments.

[Page 12, Line 278-287].

We have also added a label to all the figures to clarify the definition of the dependent variable on the y-axis as follows: 

*Note. Accuracy (%) reflects the degree to which responses correspond with the set of predetermined mental-states of the characters in the Vignettes a priori.

REVIEWER #3's COMMENTS

Comment 1. The authors examine how adults use different piece of information to infer mental states. I find the idea of studying how people incorporate all sorts of information into their mental state reasoning fascinating because I agree with the authors that researchers often narrow their focus to one specific cause of a mental state so we know little about how people actually use multiple piece of (sometimes conflicting) information. Despite my excitement about this idea, my overall evaluation of the paper is less positive. In particular, I feel that the scope of the paper is less than promised by the Introduction. I also have some theoretical and methodological concerns.

Theoretically, I think it will be important for the authors to consider the trait attribution literature. Some discussion of why we would expect people to use inconsistent versus consistent information to be the same versus different across mental states and traits is needed. This feels especially necessary because the participants are actually introduced to a person with “mental states” over several points in time which to me feels more trait-like (i.e., what is this person’s stable attitudes?) than mental-state like (i.e., what is this person feeling in the here-and-now?). As it stands, I am not clear what we have learned about how people process consistent and inconsistent information about a person over time.

Response 1. We appreciate the reviewer’s insightful perspective on this matter. We have now added a paragraph in the limitations section of the study to address this: 

Furthermore, it is unclear whether participants are making inferences about mental states or traits. Given that participants are introduced to a character with “mental states” over several points in time, it remains unclear whether attributions about the individual are “state-level” attributions or “trait-level” attributions. While this is an important consideration, ToM abilities themselves are thought to influence conceptualizations of traits [reference included]. Indeed, mental states are often related to dispositional traits [reference included] and likely underly cognitive mechanisms that facilitate the understanding that traits can be causally linked with desires [reference included]. Nevertheless, follow-up questions that index whether participants believe the statement is true of the individual in that particular moment (state) versus an aspect of the individual’s character (trait) could help clarify the distinction.

[Page 33, Line 806-815].

Comment 2.

The authors appear especially interested in how uncertainty shapes people’s ability to engage in mental state reasoning. I think many of their claims about “uncertainty” are dependent on whether the participants actually felt uncertain. I would recommend that the researchers run similar studies where they actually measure whether participants felt more or less certain across conditions. The authors acknowledge this in their Discussion, but given that they are interested in uncertainty, I think data is needed to speak to this concern. Alternatively, I see that the authors have reaction time data, they could potentially use this to speak to uncertainty.

Response 2. We would like to thank this reviewer for the great suggestion! As recommended, we were able to analyze reaction time (RT) data (as an indirect measure of uncertainty). Essentially, if participants have longer reaction times, then it is possible that they have higher levels of uncertainty. In the reaction time data analysis reported in S3 Appendix, the results showed longer reaction times in the high-uncertainty condition compared to the low- and intermediate-uncertainty conditions in Experiment 1, longer reaction times in the low-quantity condition compared to the intermediate- and high-quantity conditions in Experiment 2, and shorter reaction times in the high-consistency condition compared to the low- and intermediate-consistency conditions in Experiment 3. Such results indicate that our manipulations are influencing participants’ reaction times, which may reflect the underlying uncertainty when making inferences. We have now added the following sentences in the Limitations and Future Directions section to briefly mention the results of reaction times [added sentences are in darker font here for illustrative purposes]:

Additionally, there is an underlying assumption that the manipulations in all three experiments are directly exerting an effect on participant’s uncertainty levels, but other measures will be necessary to confirm this assumption. When analyzing participants’ reaction times, as an indirect measure of uncertainty, the results showed that reaction times significantly differed by condition in the three experiments (see S6 Appendix for the reaction time analyses). Therefore, these results suggest that the manipulations influenced participants’ uncertainty levels when making mental-state inferences. However, more direct measures, for example, having follow-up questions with a certainty index immediately after participants viewed each vignette could provide converging evidence for how uncertainty factors into mental state computation.

[Pages 33, Line 798-802].

Comment 3.

I would recommend not using the term “accuracy” but rather just saying that the date were coded in line with the idea that Lewis likes Clara’s piano playing. To me, I find it confusing to say that the “high-uncertainty” condition is less accurate than the “low-uncertainty” condition, it more seems like they have different interpretations because they have different information. I certainly wouldn’t want to say that the participants in the “high-uncertainty” condition were worse mental-state reasoners than those in the “low-uncertainty” condition.

Response 3. We would like to thank all the reviewers for their insightful comments and feedback. This comment is highly relevant and is an integral element of our paper. It was also noted by Reviewer 2. The fact that the same inquiries are being posited by several blind reviewers is indicative of the importance of clarifying the definition of the dependent variable. To satisfy both reviewers, we have addressed this comment by adding a detailed description of the dependent variable to ensure that the reader has a clear understanding of what the term refers to. We have also revised all figures to include a label defining the dependent variable depicted on the y-axis. For further details on how we addressed this comment, please see our response to Reviewer 2’s Comment #1.

Comment 4.

The authors use many different words throughout to (I think) mean the same thing (consistency, type, quality), this makes the arguments and conditions of the paper a little difficult to follow.

Response 4. We have addressed this excellent comment by rewording the labels across the entire manuscript to indicate that the uncertainty manipulations are on the amount of information and the consistency of information, to avoid any confusion for the reader. For further details on how we added this comment, please see our response to Reviewer 1’s Comment #4. 

Comment 5.

I was surprised to see that the authors randomized the order that the different information came in given that prior research suggests that the order people learn information matters (e.g., Cone, Flaharty, & Ferguson, 2021). Is it possible for the authors to explore differences in people’s mental state attributions by order of information?

Response 5. We agree completely with this reviewer that condition order is an important variable, and we would like to thank them for this insightful and thought-provoking comment. After deep analysis of the rationale for randomizing the condition order, we would like to note that the reason why this was done was precisely because condition order may be an influential variable on the outcome. To control for this and other potential recency or primacy memory effects, we had randomized the order in which statements appear. To address this comment, we have cited this pertinent paper and have since added the following to the Design section of Experiment 1, Paragraph 3 (please note that for Experiments 2 and 3, the same logic was used). 

Condition order and sentence order were randomized; this is because there is evidence that condition order may be an important variable[reference included], thus in the present set of experiments, we controlled for condition order by randomizing the order of sentences. Specifically, the location of each sentence (i.e., a filler sentence, a consistent sentence or an inconsistent sentence) within the block of text for each vignette was random.

[Pages 10-11, Line 244-257].

We also agree with this reviewer that this would be a very interesting avenue for future research, as the order seems to matter in participants' inferences. However, discerning the effects of order on mental state attributions was not the central aim of the present study and we encourage future research to explore this further. We have added the following statement to the Limitations and Future Directions section to emphasize this:

Future research can also explore the extent to which the order in which various statements appear influence mental state attributions.

[Page 34, Line 842-843].

Comment 6.

I am a little confused about a couple of aspects of the design. First, when the authors say that the order of conditions was counterbalanced, does this mean that they were blocked (so, for example, all of the intermediate, then all of the low-uncertainty, and then all of the high-uncertainty) or were all of the stories counterbalanced so that participants might get an intermediate story, and then a low-uncertainty, and then another intermediate story, and then a high-uncertainty, etc.? If the stories were blocked, please include an order effect analysis. This would be especially important if the authors want to use the reaction time data as evidence of uncertainty because it's possible that people's certainty went up or down based on what condition they saw first.

Response 6. We would like to thank this reviewer for their detailed and constructive analyses of the randomization technique for creating the stimuli. To answer the highly pertinent question raised, we did not block the order of conditions. A participant could view the first vignette and by chance, it could be a low-uncertainty condition, or an intermediate-uncertainty condition, or a high-uncertainty condition. Each vignette shown was randomly selected to avoid any possible order effects. We agree that order effects of primacy and recency can have significant influences on memory and reasoning and thus, we randomized as many elements of the stimuli as possible. We have since clarified this by removing the statement that conditions were “counterbalanced” and instead, clarifying that they are simply randomized. We have also added a section following the Design section of Experiment 1, Paragraph 3 (please note that for Experiments 2 and 3, the same logic was used). This added level of description adds further clarity for the reader and thus, we appreciate this reviewer’s attention to detail.

[...] whether or not the first vignette that a participant sees is an intermediate-uncertainty, low-uncertainty, or high-uncertainty condition was also randomized to ensure that we have controlled for all possible order effects.

[Page 11, Line 257-259].

Comment 7.

I also wasn’t sure if the 6 different “consistent” statements from the “low-uncertainty” condition could be randomly slotted into each of the “consistent” statements for the intermediate and high-uncertainty conditions or whether which “consistent” statements were used in the intermediate and high-uncertainty conditions were fixed across participants. This seems important to know especially because it does not appear that the authors have pilot tested these statements to ensure that they all provide similar information.

Response 7. Reviewer 3 has rightly pointed out an important element of the vignettes. To answer the question posed, of the 6 different “consistent” sentences, the selected sentences chosen to appear in the vignettes of the other conditions were also random. This was because it may have been the case that some statements are more potent and convincing in their content than others so to control for any possible effects of this and there may be individual differences in which statement participants would feel convey more potent information on mental states, we randomly chose statements to be slotted into the other conditions. To address this comment, we have added the following at the end of the Design section of Experiment 1, Paragraph 3 (please note that for Experiments 2 and 3, the same logic was used). 

And finally, it is possible that some statements may be more potent and convincing in their content than others. For example, of the 6 "consistent" sentences, although all attempt to convey the notion that Lewis likes Clara's piano playing, it is possible that some may be stronger in portraying this mental state attribution than others and that there are individual differences in which statements participant feel convey more potent information on mental states. To control for this, the select sentences chosen from the 6 consistent sentences in the low-uncertainty condition to be inserted in the other conditions (intermediate-uncertainty and high-uncertainty) were randomly selected.

[Page 11, Line 259-266].

 Comment 8.

It would be helpful if the authors could provide all of the vignettes that were used. Was it the case that the “correct” mental state was always positive (e.g., Lewis like’s Clara’s piano playing) or was it sometimes the case that the person would have a negative mental state towards the person? Relatedly, were the characters’ mental states always social (i.e., what they thought of another person) or were they sometimes asocial (e.g., their belief about where an object is). These are important pieces of information to report in the manuscript.

Response 8. We would like to thank this reviewer for requesting the stimuli. We have since created a supplementary attachment with all the vignettes “S1 Appendix Examples of Vignette Scripts” for readers to refer to. To answer the inquires posited, the characters’ mental states were not necessarily always social; the element onto which the character would have a set of mental states about could be a person, place, thing, or phenomenon and this was decided using a random number generator. To answer this reviewer's comment about the valence of the mental state attributions, they were not necessarily always positive and could have been negative (i.e., that a character does not like something). We have included these details in the supplementary material (“S2 Appendix Randomization Technique for Vignettes”) in which all details pertaining to the creation of the stimuli (including the important elements raised by this reviewer) are addressed. 

Comment 9.

The authors argue that the findings from Experiment 1 and 2 are similar. I’m not sure I agree with this. Yes, they get the same pattern, but the effect seems much larger in Experiment 1. This should be addressed.

Response 9. We agree with the reviewer that this interesting finding needs to be addressed more clearly in the manuscript. We have therefore added a section in the General Discussion (paragraph 4) of the revised manuscript to describe the main findings and the fact that while the patterns are similar, the effect is much larger when the consistency of information is altered. 

While the findings from all three Experiments show the same pattern, some important differences are noteworthy. The findings of Experiment 1 (in which both information amount and information consistency were altered) are strong, but after teasing out the potential effects of the amount and consistency of information in Experiment 2 (amount) and Experiment 3 (consistency), an interesting set of results emerge. Namely, the effect observed in Experiment 2 is not as large as that in Experiment 1, suggesting that the inclusion of the consistency manipulation is influencing mental state attributions. And furthermore, comparing Experiment 2 with Experiment 3, it is once again clear that the result is showing the same pattern, but the effect is much larger in Experiment 3 in which only the consistency of information was manipulated. Such patterns imply the significance of consistency to reason about others’ mental states under uncertainty.

[Page 30, Line 711-721].

---

## [Editor Report · Decision Letter 1]

6 Sep 2022

PONE-D-22-13954R1Reasoning About Mental States Under UncertaintyPLOS ONE

Dear Dr. Kamkar,

Thank you for submitting your manuscript to PLOS ONE. After careful consideration, we feel that it has merit but does not fully meet PLOS ONE’s publication criteria as it currently stands. Therefore, we invite you to submit a revised version of the manuscript that addresses the points raised during the review process.

You have addressed to all of the reviewers' point. However, the response to Reviewer 2's first and "primary" concern is not sufficiently convincing. That point is absolutely crucial for the interpretation of the study results. A way to disentangle the issue about what the "correct" answer actually is, and how participants actually weigh the available pieces of information, is to quantify them. For now, you only have an ordinal classification (low, intermediate, high) based on your own judgement alone. You might collect some additional rating data from naive participants judging not only if, but also how much each single piece of information is consistent and indicative of specific states of minds. In this way, you can quantify, for each single trial, how much and how consistent the available information is.

A related point concerns the data analysis. Once you have quantified the amount and consistency of information available for each trial and you can use this information as predictors, you should analyze the data using statistical methods more appropriate than Anovas with normality assumptions. Since the actual responses are binomial (correct/incorrect) and they are repeated by participants and by trial/scenario, using mixed-effects logistic regressions is recommended. (This is recommended even if you do not quantify amount and consistency of information and you keep the independent variable as it is now.)

We look forward to receiving your revised manuscript.

Kind regards,

Enrico Toffalini, Ph.D

Academic Editor

PLOS ONE

Additional Editor Comments:

You have addressed to all of the reviewers' point. However, the response to Reviewer 2's first and "primary" concern is not sufficiently convincing. That point is absolutely crucial for the interpretation of the study results. A way to disentangle the issue about what the "correct" answer actually is, and how participants actually weigh the available pieces of information, is to quantify them. For now, you only have an ordinal classification (low, intermediate, high) based on your own judgement alone. You might collect some additional rating data from naive participants judging not only if, but also how much each single piece of information is consistent and indicative of specific states of minds. In this way, you can quantify, for each single trial, how much and how consistent the available information is.

A related point concerns the data analysis. Once you have quantified the amount and consistency of information available for each trial and you can use this information as predictors, you should analyze the data using statistical methods more appropriate than Anovas with normality assumptions. Since the actual responses are binomial (correct/incorrect) and they are repeated by participants and by trial/scenario, using mixed-effects logistic regressions is recommended. (This is recommended even if you do not quantify amount and consistency of information and you keep the independent variable as it is now.)

---

## [Author Response · Author response to Decision Letter 1]

23 Sep 2022

RESPONSE TO REVISIONS

1. Reviewer 2 Comment: The definition of “accurate” answers

"You have addressed to all of the reviewers' point. However, the response to Reviewer 2's first and "primary" concern is not sufficiently convincing. That point is absolutely crucial for the interpretation of the study results. A way to disentangle the issue about what the "correct" answer actually is, and how participants actually weigh the available pieces of information, is to quantify them."

Response 1. We agree completely with this comment and have made an important revision to the terminology of the dependent variable in order to address this point more thoroughly. In terms of the definition of “accurate” answers, we had detailed in the manuscript that we predefined accurate responses as those corresponding to the answers when only consistent information was provided. Therefore, as Reviewer 2 pointed out, in the high-uncertainty condition (4 inconsistent statements and 2 consistent statements), participants’ reasonable answers should be the answers corresponding to inconsistent information. Considering our definition of accurate answers mentioned earlier, participants’ responses in the high-uncertainty condition does not mean that their responses were “wrong" or "incorrect.” It reflects the extent to which their responses correspond to the predetermined set of mental states of the character in the vignette. Reviewer 2’s point is highly pertinent and in line with our definition of the dependent variable. Although we added a considerable amount of detail and explanation of this outcome in our original set of revisions, upon further reflection and in light of this round of re-revision, we have changed the term "accuracy" to "congruency score" to more closely reflect that the score is indicative of how congruent the response is to our predetermined set of mental states for the character. This way, a low congruency score does not convey the notion that participants are scoring poorly on mental state reasoning, but that their score is simply less congruent with our preset mental states. We feel that this change in terminology conveys a more neutral definition of participant scores than the previous term "accuracy", which implied that participants who score lower are inaccurate or poor in mental stat reasoning abilities. We would once again like to thank the Editor and Reviewer 2 for initiating this important revision. We have added the following:

Congruent scores were specified as those corresponding to the answers when only consistent information was provided. For example, at the outset of the trial, the experimenters pre-set a high "congruency score" to reflect scores indicating that the character Lewis likes Clara's piano playing. Thus, lower congruency scores in the high uncertainty condition do not indicate poor ToM ability, but rather, that participants’ responses reflected the information provided, which is inconsistent with the pre-set “congruent” response. 

Location: Page 11, Line 226-231

["Accuracy" term renamed to the more appropriate term, "Congruency Score"]

Locations: Throughout Manuscript, Figure Y-Axis Labels, & S2 Appendix "Generation of Congruency Score" Section

2. Editor and Reviewer 2’s Comment: How participants actually weight the pieces of information

"For now, you only have an ordinal classification (low, intermediate, high) based on your own judgement alone. You might collect some additional rating data from naive participants judging not only if, but also how much each single piece of information is consistent and indicative of specific states of minds. In this way, you can quantify, for each single trial, how much and how consistent the available information is."

Response 2. Another point that the Editor and Reviewer 2 raised is with regards to the stimuli used in the experiments; the suggestion was to collect more data to ensure whether (a) each statement is genuinely consistent (or inconsistent) with the preset mental states of the character and (b) participants weightings of each of the pieces of information in each vignette to determine how informationally rich each statement is. To help illustrate this excellent comment, we've pasted below a sample vignette used in the study for reference.

Sample Vignette from Experiment 1

[Introduction:] Lydia is a dental assistant. Lydia recently got engaged to Andrew, her high-school sweetheart. 

[Consistent 1.] Most of the time, Lydia keeps the temperature in her and Andrew’s house relatively cool, at about 10 degrees Celsius (or 50 degrees Fahrenheit). 

[Consistent 2.] In the dental office, Lydia’s colleagues constantly complain about how cold she sets the temperature. 

[Consistent 3.] Even in the winter, Lydia often sleeps without a blanket. 

[Consistent 4.] When discussing possible honeymoon locations, Lydia suggested that she and Andrew consider a winter resort. 

[Consistent 5.] One day at work, Lydia came across an advertisement for Jamaica as a honeymoon destination, she looked up the average weather in Jamaica. After learning that the temperature in Jamaica during their honeymoon will reach as high as 33 degrees Celsius (91 degrees Fahrenheit), Lydia raised her eyebrows. 

[Consistent 6.] With a large red marker, Lydia drew a large “X” on the ad for Jamaica. 

For example, "Consistent Statement #4" can be more informationally rich than "Consistent Statement #2" in making the assertion that Lydia will not choose Jamaica for their honeymoon. 

Regarding point (a), we completely agree that additional data collection (i.e., asking naïve participants to rate whether each statement is consistent or inconsistent) would be helpful as a stimulus manipulation check, but unfortunately, we do not currently have the resources to collect more data. The resources required for such a feat include the availability of independently initiating/continuing research ethics board approval as well as funds to conduct another series of 3 follow-up experiments. We have thought critically and reflected carefully to explore our available options to address this. Considering our limited resources and feasibility constraints, a reasonable solution is to include a detailed subsection in the limitations section of the discussion devoted to discussing this transparently. We have also included a corresponding section in the future directions section with the trial outlined in detail as an avenue for future research. Moreover, our stimuli are submitted with our manuscript, allowing future researchers the means to conduct these experiments. However, please note that (even without the further data collection for manipulation check) the current data still show that the amount and consistency of information influence participants’ uncertainty and decisions regarding another person’s mental states. The added sections are as follows, with bolded sections indicating subheadings in the manuscript:

Limitations and Future Directions

[...]

Importantly, the stimuli and sentences themselves are wanting of a manipulation check to ensure that they are, in fact, consistent (or inconsistent) and if so, how much each sentence contributes to participants' mental state inferences. A follow-up study that further investigates whether and how participants weigh the available pieces of information will be necessary to clearly address the limitation. For example, using a rating design in which asking participants to quantify how much each single piece of information is consistent will elaborate the current findings. In this way, one could quantify, for each single trial, how much and how consistent the available information is.

Location: Page 33, Line 713-720

Regarding point (b), we were impressed with your great suggestion to design the experiment to quantify how helpful each statement in the vignettes is in cuing participants to their mental state inference. Based on the current task design and dataset, we have minimized the possible effect of different richness levels of each statement on the manipulation through the randomization process. Moreover, with 45 separate vignettes each consisting of 6 sentences, the likelihood that a particular sentence is driving any effects is low. Having said that, we agree that your suggestion of more data collection with a rating design can clearly ensure the consistency levels of each story, which is why we added these details to the Limitations and Future Directions sections (as noted above). We feel that these additions provide the reader with a deeper appreciation for our task, while being aware of its limitations and future experiments that can help resolve and improve its utility. This comment provided us with an opportunity to reflect on the virtues and merits of our stimuli, which we have now added as follows:

Despite the limitations outlined in the stimuli used within the three experiments of the current study, there are several features of the stimuli that are noteworthy. Contrary to other existing ToM tasks in which one definite correct answer exists (e.g., Sally-Anne task in which participants are asked to infer another agent’s mental state on an object’s location; location A or B?), the type of ToM tasks used in the current study (i.e., participants are asked to infer the mental state of others on their attitudes, beliefs, and emotions towards something or someone, which can be mixed or multiple), there are a plethora of opportunities to infer mental states with multiple questions targeting various mental state processes. This process provided us with a richer response for our primary outcome (i.e., rather than "location A/B" in previous ToM tasks) and is more reflective of mental state reasoning in every day, real life processes. That is, we typically do not simply ask ourselves a binary question pertaining to another individual's mental states. But rather, we integrate the complexities of people's mental states with a series of inferences on what the other person thinks, how they feel, what they believe to be true, and how they will likely behave in the future. We are also often faced with competing and complex information that may alter our mental state inferences, features which our task and manipulations capture. Furthermore, with our task, there can always be individual differences in which statement can be perceived as more rich to infer others’ mental states. To address this in the current set of studies and rule out any individual difference factors, we randomized many aspects of the stimuli including which sentences were chosen to appear in the vignettes of the other conditions. Therefore, it is unlikely that one informationally rich statement drove any effects observed, as the likelihood of each sentence being chosen in other conditions was random, as was the likelihood of a participant being exposed to each experimental condition. 

Location: Pages 34-35, Lines 742-763.

3. Data Analysis Suggestion

"A related point concerns the data analysis. Once you have quantified the amount and consistency of information available for each trial and you can use this information as predictors, you should analyze the data using statistical methods more appropriate than Anovas with normality assumptions. Since the actual responses are binomial (correct/incorrect) and they are repeated by participants and by trial/scenario, using mixed-effects logistic regressions is recommended. (This is recommended even if you do not quantify amount and consistency of information and you keep the independent variable as it is now.)"

Response 3. And finally, conducting a binomial mixed-effects logistic regression to analyze the data was suggested. We agree that this suggestion would be most appropriate with binomial data, but our current accuracy measure is continuous. We have added the following to clarify that our dependent variable is operationalized as a continuous variable in the manuscript (bolded statement indicates new additions for illustrative purposes in this letter).

To address the hypothesis of a direct effect of the uncertainty manipulation on responses to questions (as a continuous dependent variable) pertaining to the agent's mental states, a repeated-measures ANOVA is required (with follow-up pairwise tests if applicable).

Location: Page 7, Line 158-161.

---

## [Decision Letter · Decision Letter 2]

26 Oct 2022

Reasoning About Mental States Under Uncertainty

PONE-D-22-13954R2

Dear Dr. Kamkar,

We’re pleased to inform you that your manuscript has been judged scientifically suitable for publication and will be formally accepted for publication once it meets all outstanding technical requirements.

Kind regards,

Enrico Toffalini, Ph.D

Academic Editor

PLOS ONE

Additional Editor Comments (optional):

Reviewers' comments:

Reviewer's Responses to Questions

**Comments to the Author**

1. If the authors have adequately addressed your comments raised in a previous round of review and you feel that this manuscript is now acceptable for publication, you may indicate that here to bypass the “Comments to the Author” section, enter your conflict of interest statement in the “Confidential to Editor” section, and submit your "Accept" recommendation.

Reviewer #2: All comments have been addressed

Reviewer #3: All comments have been addressed

2. Is the manuscript technically sound, and do the data support the conclusions?

Reviewer #2: Yes

Reviewer #3: Yes

3. Has the statistical analysis been performed appropriately and rigorously? 

Reviewer #2: Yes

Reviewer #3: Yes

4. Have the authors made all data underlying the findings in their manuscript fully available?

Reviewer #2: Yes

Reviewer #3: Yes

5. Is the manuscript presented in an intelligible fashion and written in standard English?

Reviewer #2: Yes

Reviewer #3: Yes

6. Review Comments to the Author

Reviewer #2: The authors have largely addressed the remaining concerns. Though additional data to evaluate the extent to which each bit of information is consistent, and how that information is weighed, would be very helpful towards evaluating the findings, I'm sensitive to the fact that additional data collection, due to limited resources, is not always possible as seems to be the case here. I also think the editor's suggestion to conduct mixed-effects logistic regressions that evaluates trial-by-trial responses is a good one. But in theory, the data are available from the authors if someone wanted to run that analysis.

Reviewer #3: I thank the authors for addressing my concerns and for raising the important limitations that still exist from this work.

7. PLOS authors have the option to publish the peer review history of their article (what does this mean?). If published, this will include your full peer review and any attached files.

Reviewer #2: No

Reviewer #3: No
